# Learning nonequilibrium statistical mechanics and dynamical phase transitions

Ying Tang [1,2,8] ✉, Jing Liu [3,4,8], Jiang Zhang [4,5] & Pan Zhang [3,6,7] ✉

Nonequilibrium statistical mechanics exhibit a variety of complex phenomena far from equilibrium. It inherits challenges of equilibrium, including accurately describing the joint distribution of a large number of configurations, and also poses new challenges as the distribution evolves over time. Characterizing dynamical phase transitions as an emergent behavior further requires tracking nonequilibrium systems under a control parameter. While a number of methods have been proposed, such as tensor networks for one-dimensional lattices, we lack a method for arbitrary time beyond the steady state and for higher dimensions. Here, we develop a general computational framework to study the time evolution of nonequilibrium systems in statistical mechanics by leveraging variational autoregressive networks, which offer an efficient computation on the dynamical partition function, a central quantity for discovering the phase transition. We apply the approach to prototype models of nonequilibrium statistical mechanics, including the kinetically constrained models of structural glasses up to three dimensions. The approach uncovers the active-inactive phase transition of spin flips, the dynamical phase diagram, as well as new scaling relations. The result highlights the potential of machine learning dynamical phase transitions in nonequilibrium systems.

Tracking time evolution and characterizing phase transitions are fundamental tasks in nonequilibrium statistical mechanics[1], having implications to a wide range of fields including quantum transport[2], molecular machines[3], quantitative biology[4,5], and complex networks[6]. For example, the glassy behavior has been explored in kinetically constrained models (KCM) of spin flips on a lattice[7], where each spin can facilitate the flip of its neighbor spins. The stochastic dynamics of spin flips give insights into the dynamical heterogeneity of the glass transition[8]. Another example is the voter model[9] describing consensus formation, where voters located on a network choose the opinion based on their neighbors and can form characteristic spatial structures.

Despite tremendous efforts, studying the dynamical phase transition remains challenging in general. First, it requires tracking the evolving probability distribution of microscopic configurations, which is exponentially increasing with the system size and can be computationally prohibitive from sampling trajectories[10]. Second, the phase transition is induced by a control parameter, and the corresponding dynamical operator governing the time evolution, known as the "tilted" generator[11], does not preserve the normalization condition of the distribution. Thus, one needs to estimate the normalization factor, i.e., the dynamical partition function, a central quantity in nonequilibrium systems[12]. Third, studying nonequilibrium dynamics at an arbitrary time is even more challenging, because it demands estimating the whole spectrum of the tilted generator, rather than computing only the largest eigenvalue in the long-time limit[13,14] based on the large deviation theory[11,15]. Analytically, it is intractable except for rare

[1]Institute of Fundamental and Frontier Sciences, University of Electronic Sciences and Technology of China, Chengdu 611731, China. [2]International Academic Center of Complex Systems, Beijing Normal University, Zhuhai 519087, China. [3]CAS Key Laboratory for Theoretical Physics, Institute of Theoretical Physics, Chinese Academy of Sciences, Beijing 100190, China. [4]School of Systems Science, Beijing Normal University, Beijing 100875, China. [5]Swarma Research, Beijing 102308, China. [6]School of Fundamental Physics and Mathematical Sciences, Hangzhou Institute for Advanced Study, UCAS, Hangzhou 310024, China. [7]Hefei National Laboratory, Hefei 230088, China. [8]These authors contributed equally: Ying Tang, Jing Liu. ✉e-mail: jamestang23@gmail.com; panzhang@itp.ac.cn

cases[16–18]; numerically, the major difficulty of studying time evolution beyond the steady state arises from expressing the high-dimensional probability distribution with a growing complexity when the correlation builds up over time.

In this work, we develop a general framework to track nonequilibrium systems over time by variational autoregressive networks (VAN), which offers an ideal model for describing the normalized joint distribution of configurations[19]. The VAN was previously shown effective to investigate equilibrium statistical mechanics[20–22], quantum many-body systems[23–25], chemical reaction networks[26] and computational biology[27]. However, evaluating the dynamical partition function and characterizing dynamical phase transitions have not been achieved in these applications of the VAN. Here, we leverage the VAN to propose an algorithm for tracking the evolving probability distribution, leading to an efficient computation of the dynamical partition function. The latter serves as the moment-generating function of dynamical observables, which uncovers the phase transition over time (Fig. 1).

We apply the approach to representative models in nonequilibrium statistical mechanics. We first validate the method in the voter model, by comparing with the analytical result[6]. To demonstrate that our approach can reveal unknown dynamical phenomena, we investigate KCMs of spin flips for glassy dynamics, including the Fredrickson-Andersen (FA) model[28] and variants of the East model[29], in one dimension (1D), two dimensions (2D), and three dimensions (3D). Previously, the dynamical active-inactive phase transition in space and time[30–32] was investigated mainly at the steady state in the long-time limit, where KCMs have two phases with extensive or subextensive spin-flipping activities. However, the phase transition in the full-time regime was seldom investigated. For the 1D finite-time problem, our results agree with the recent study by matrix product states[33]. For 2D and 3D cases, where the phase transition at arbitrary time was not obtained either analytically or numerically, our method uncovers the dynamical phase diagram versus time the control parameter of the counting field, and the critical exponent of the finite-time scaling. We also observe the emergence of characteristic spatial structures over

time, extending the steady-state result[34]. We further discuss the applications to nonequilibrium systems with other types of dynamics and topologies.

## Results

### Nonequilibrium statistical mechanics

We consider a continuous-time discrete-state Markovian dynamics of size $N$. Each variable has $d_s$ states ($d_s = 2$ for binary spin systems), giving total $M = d_s^N$ configurations. With configuration states $|\mathbf{x}\rangle \equiv (x_1, x_2, \ldots, x_N)$ forming an orthonormal basis, the system at time $t$ is described by the probability vector $|P_t\rangle = \sum_{\mathbf{x}} P_t(\mathbf{x})|\mathbf{x}\rangle$. It evolves under the stochastic master equation[35]:

$$\frac{d}{dt}|P_t\rangle = \mathbb{W}|P_t\rangle, \tag{1}$$

where $\mathbb{W} = \sum_{\mathbf{x}, \mathbf{x}' \neq \mathbf{x}} w_{\mathbf{x}, \mathbf{x}'}|\mathbf{x}'\rangle\langle\mathbf{x}| - \sum_{\mathbf{x}} r_{\mathbf{x}}|\mathbf{x}\rangle\langle\mathbf{x}|$ is the generator, $w_{\mathbf{x}, \mathbf{x}'}$ is the transition rate from $|\mathbf{x}\rangle$ to $|\mathbf{x}'\rangle$, and $r_{\mathbf{x}} = \sum_{\mathbf{x}' \neq \mathbf{x}} w_{\mathbf{x}, \mathbf{x}'}$ is the escape rate from $|\mathbf{x}\rangle$.

**Dynamical partition function.** Studying dynamical phase transitions emerging from microscopic interactions relies on evaluating dynamical observables. In general, a quantity of interest is the time-extensive dynamical observable $\hat{K}$ incremented along a trajectory $\omega_t = \{\mathbf{x}_{t_0} \to \mathbf{x}_{t_1} \cdots \to \mathbf{x}_t\}$, with time-step length $\delta t$, $t = J\delta t$ and total $J$ time steps. The probability of observing the dynamical observable with a value $K$ is obtained by summing over all possible trajectories: $P_t(K) = \sum_{\omega_t} p(\omega_t)\delta[\hat{K}(\omega_t) - K]$, where $p(\omega_t)$ is the probability of the trajectory $\omega_t$.

To extract the statistics of the dynamical observable, it is useful to estimate its moment-generating function, i.e., the dynamical partition function $Z_t(s) = \sum_K P_t(K)e^{-sK} = \sum_{\omega_t} p(\omega_t)e^{-sK(\omega_t)}$, where a control parameter $s$ is introduced as the conjugate variable to the dynamical observable. Taking derivatives of $Z_t(s)$ to $s$ gives moments of the dynamical observable's distribution. The dynamical partition function

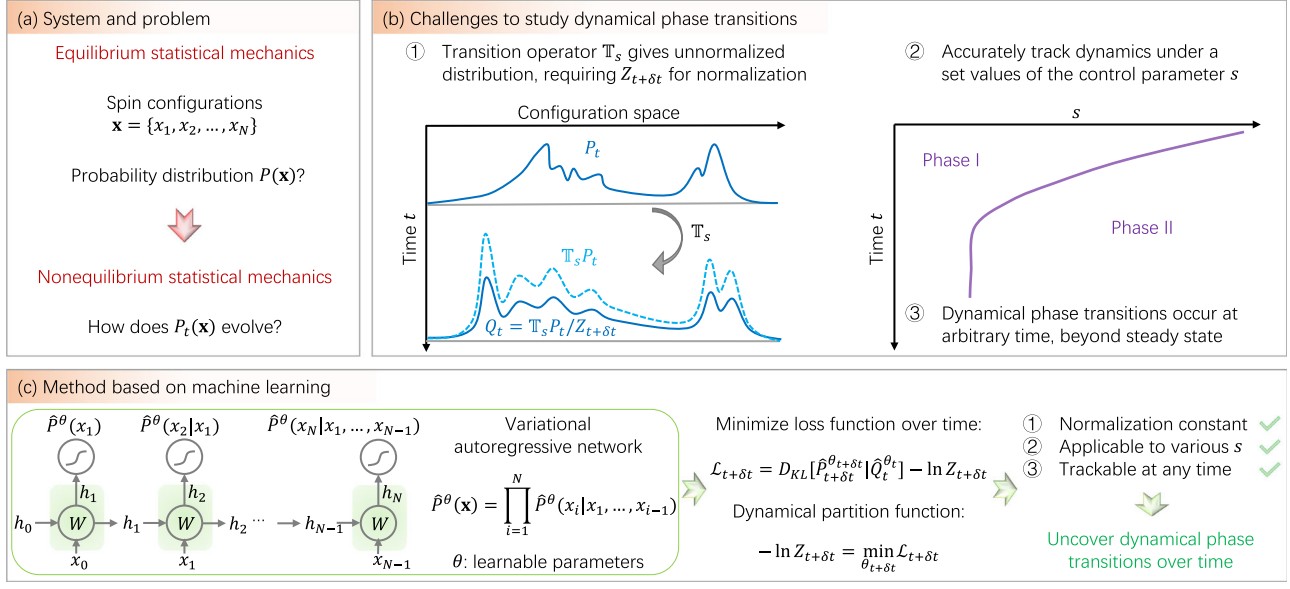

**Fig. 1 | Uncovering dynamical phase transitions in nonequilibrium statistical mechanics by machine learning. a** The major theme of equilibrium and nonequilibrium statistical mechanics. **b** Uncovering dynamical phase transitions requires studying the "tilted" dynamics under $\mathbb{T}_s$ with various values of the control parameter $s$, where the evolved distribution is no longer normalized (dashed light blue). As the normalization factors, the dynamical partition function needs to be learned over time, to discover the dynamical phase diagram. **c** A proposed

algorithm for learning the dynamical partition function by training the VAN (distributions with hat) over time. The loss function is given by the KL-divergence between the VAN at time $t + \delta t$ with learnable parameters $\theta_{t+\delta t}$ and the evolved distribution learned at $t$ with parameters $\theta_t$ fixed. The algorithm tracks nonequilibrium dynamics, and reveals dynamical phase transitions, overcoming the three challenges in (**b**).

can be evaluated by[30]:

$$Z_t(s) = \langle -|e^{t\mathbb{W}_s}|ss\rangle, \tag{2}$$

where $\langle -| = \sum_{\mathbf{x}}\langle \mathbf{x}|$, and $|ss\rangle$ is the steady-state probability vector under the generator $\mathbb{W}$. Different from $\mathbb{W}$, $\mathbb{W}_s$ is termed as the "tilted" generator, and its form depends on the dynamical observable $\hat{K}$ and control parameter $s$.

**A representative dynamical observable.** For the spin-flip dynamics, a characteristic dynamical observable is the "dynamical activity"[30,36], measuring the number of spin flips. The observable quantifies how dynamically active the trajectories are: a more active trajectory has a higher value. The control parameter $s$ is then termed as the "counting field". To evaluate the dynamical observable, the generator is modified correspondingly as the tilted generator[30]:

$$\mathbb{W}_s = \sum_{\mathbf{x}, \mathbf{x}' \neq \mathbf{x}} e^{-s} w_{\mathbf{x}, \mathbf{x}'}|\mathbf{x}'\rangle\langle \mathbf{x}| - \sum_{\mathbf{x}} r_{\mathbf{x}}|\mathbf{x}\rangle\langle \mathbf{x}|, \tag{3}$$

under which the probability is no longer normalized. Combining Eq. (2) and Eq. (3), the moments can be calculated, including the average dynamical activity per unit of time and site:

$$k_t(s) = -\frac{1}{Nt}\frac{d}{ds}\ln Z_t(s). \tag{4}$$

Other dynamical observables can be investigated in a similar way by using its corresponding tilted generator.

## Tracking nonequilibrium statistical mechanics and dynamical phase transitions

A natural approach of studying nonequilibrium dynamics is tracking the evolution equation, Eqs. (1) and (2). Unfortunately, the exact representation of $|P_t\rangle$ requires a computational effort that is exponential in the system size. Hence we need an efficient method to approximately represent $|P_t\rangle$. Here, we consider a neural-network model, the VAN[19], as a variational ansatz for $|P_t\rangle$. The VAN maps configurations to the probability distribution $P_t(\mathbf{x})$ in $|P_t\rangle$ as the product of conditional probabilities:

$$\hat{P}_t^{\theta}(\mathbf{x}) = \prod_{i=1}^{N} \hat{P}_t^{\theta}(x_i|x_1, \ldots, x_{i-1}), \tag{5}$$

where the hat symbol denotes the parameterization by the neural network with learnable parameters $\theta$.

For lattice systems, we start from an initial site and traverse the lattice in a predetermined order to acquire all the conditional probabilities (Methods). Each conditional probability $\hat{P}_t^{\theta}(x_i|x_1, \ldots, x_{i-1})$ is parameterized by a neural network, where the input is the sites visited earlier $\{x_{i' < i}\}$ and the output is $x_i$ associated with a probability under proper normalization[19]. The parameters of the neural network can be shared among sites[37] to increase computational efficiency. Since all the conditional probabilities are stored, one can efficiently generate samples associated with normalized probabilities, which can be used to compute quantities such as energy and entropy and to construct loss functions to update the parameters.

The VAN is capable of expressing strongly correlated distributions[20,22], including equilibrium distributions in statistical mechanics[19], steady state distributions of KCMs in nonequilibrium statistical mechanics[34] and quantum systems[21]. Here, we find it effective to learn the time-evolving distributions. The expressivity of the VAN and training time depends on the architecture, as well as the depth and width of the neural network (Supplementary Information Sect. VD). Typical neural-network architectures can

be employed, including MADE[38], PixelCNN[39,40] or RNN[41]. For example, we have used RNN for KCMs in 1D and 2D, PixelCNN for 2D, and MADE for 3D. In 1D, we find RNN more accurate than MADE. In 2D, RNN has a comparable accuracy with PixelCNN but takes a longer computational time. The VAN can be further improved by cooperating with more advanced neural network architectures and sampling techniques.

The advantages of the VAN over the tensor network model in representing $|P_t\rangle$ mainly come from its generality. The neural network ansatz can be used in systems with various topologies, in contrast to the matrix product states designed for one-dimensional systems. On the one hand, the VAN is suitable for systems with arbitrary topology without modifying the structure of VAN[19]; on the other hand, we can design the VAN to fit the topology. For example, in 2D or 3D, one can use convolutional networks[42], as in image recognition; for sparse networks, graph neural networks are efficient[43].

Based on the VAN representation, we evaluate Eq. (2) by applying the operator $e^{\delta t \mathbb{W}_s}$ sequentially at each of the total $J$ time steps: $Z_t(s) \approx \langle -|(\mathbb{T}_s)^J|ss\rangle$ (Suzuki-Trotter decomposition[44]), where the transition operator $\mathbb{T}_s = (\mathbb{I} + \delta t \mathbb{W}_s) \approx e^{\delta t \mathbb{W}_s}$, $\mathbb{I}$ denotes the identity operator. Without loss of generality, we consider the one-step evolution from a normalized probability vector $|\hat{P}_j^{\theta_j}\rangle$ at time step $j$ ($j = 0, \ldots, J-1$) with parameters $\theta_j$. Under the tilted generator, the evolved probability vector $\mathbb{T}_s|\hat{P}_j^{\theta_j}\rangle$ becomes unnormalized. Still, since the VAN provides a normalized probability vector, we can use it to represent $|\hat{P}_{j+1}^{\theta_{j+1}}\rangle$ at time step $j+1$ and approximate the normalized probability vector $|\hat{Q}_j^{\theta_j}\rangle = \mathbb{T}_s|\hat{P}_j^{\theta_j}\rangle / Z_{j+1}(s)$, by minimizing the Kullback-Leibler (KL) divergence,

$$D_{KL}\left[\hat{P}_{j+1}^{\theta_{j+1}}||\hat{Q}_j^{\theta_j}\right] = \mathcal{L}_{j+1} + \ln Z_{j+1}(s), \tag{6}$$

$$\mathcal{L}_{j+1} = \sum_{\mathbf{x}} \hat{P}_{j+1}^{\theta_{j+1}}(\mathbf{x}, s)\left[\ln \hat{P}_{j+1}^{\theta_{j+1}}(\mathbf{x}, s) - \ln \mathbb{T}_s \hat{P}_j^{\theta_j}(\mathbf{x}, s)\right]. \tag{7}$$

Minimizing the KL-divergence is equivalent to minimizing $\mathcal{L}_{j+1}$, which plays a role analogous to the variational free energy in equilibrium statistical mechanics. Since the VAN supports unbiased sampling in parallel to compute $\mathcal{L}_{j+1}$, we estimate the gradients with respect to parameters $\theta_{j+1}$ by the REINFORCE algorithm[45]:

$$\nabla_{\theta_{j+1}}\mathcal{L}_{j+1} = \sum_{\mathbf{x}} \hat{P}_{j+1}^{\theta_{j+1}}(\mathbf{x}, s)\Big\{\nabla_{\theta_{j+1}} \ln \hat{P}_{j+1}^{\theta_{j+1}}(\mathbf{x}, s) \\ \cdot \left[\ln \hat{P}_{j+1}^{\theta_{j+1}}(\mathbf{x}, s) - \ln \mathbb{T}_s \hat{P}_j^{\theta_j}(\mathbf{x}, s)\right]\Big\}, \tag{8}$$

where the summation is over the samples from the VAN.

As an essential outcome of the algorithm, the dynamical partition function is computed as a product of the normalization constants at each time step $Z_t(s) \approx \prod_{j=1}^{J} Z_j(s)$ (Supplementary Information Sect. IIA). For each normalization constant, the nonnegativity of the KL-divergence ensures that Eq. (7) provides a lower bound as:

$$\ln Z_{j+1}(s) \geq -\mathcal{L}_{j+1}. \tag{9}$$

The equality holds when the VAN faithfully learns the evolved distribution and achieves zero KL divergence. Then, to evaluate the dynamical partition function our algorithm starts from the steady state of the non-tilted generator and tracks the distribution under the tilted generator (Eq. (2)). The renormalization procedure over time points enables to extraction of the dynamical partition function under the tilted generator, beyond the algorithm of only tracking the evolving distribution under the non-tilted generator[26].

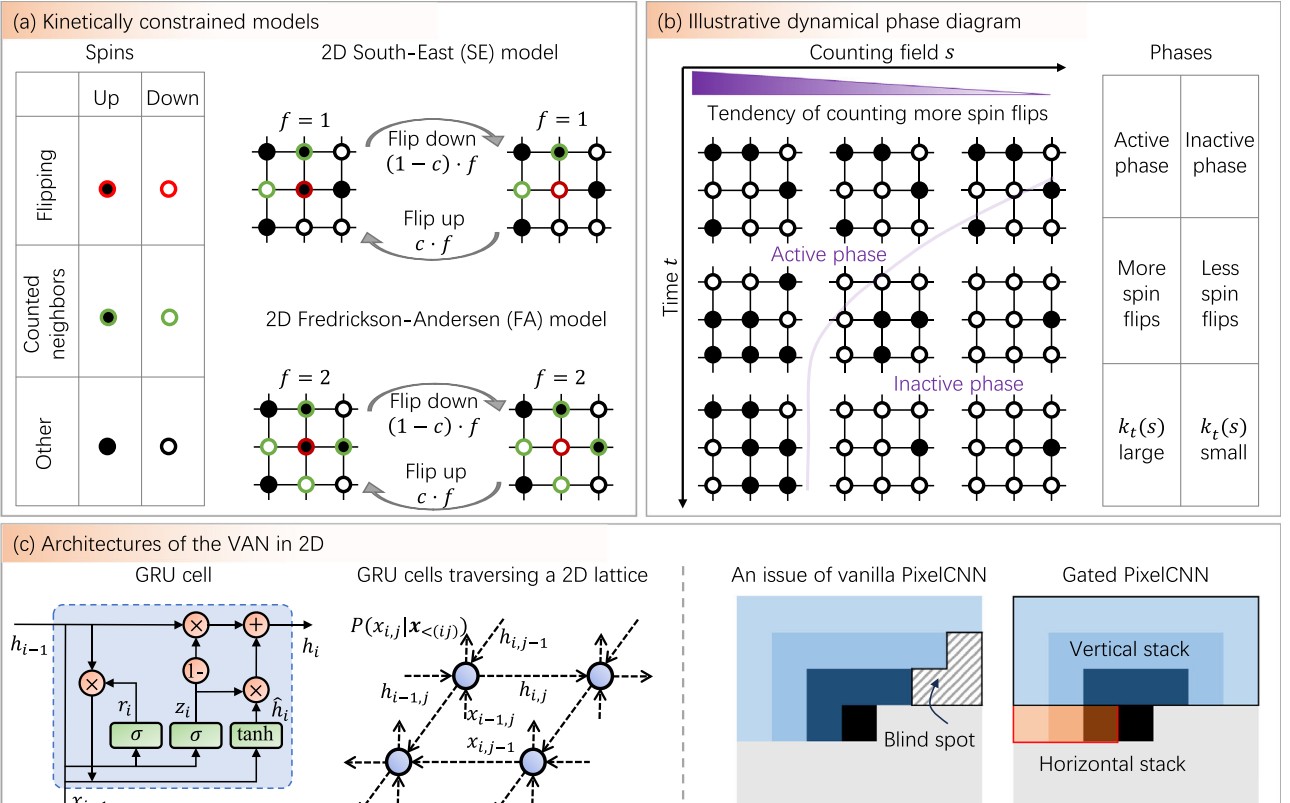

**Fig. 2 | A schematic on kinetically constrained models and dynamical phase diagram. a** The rule of spin flips for the 2D SE and FA models, as demonstrative examples. The filled (unfilled) circle is up (down) spin. The flipping rate of each spin (red) depends on $f$, the number of up spins at certain nearest neighbors (green). The SE model counts the left and above neighbors, and the FA model counts all nearest neighbors. The flip-up probability is $c$. **b** An illustrative dynamical phase diagram, with active and inactive phases of spin flips depending on the counting field $s$ and time. The active (inactive) phase has more (less) spin flips over time, giving large (small)

dynamical activity $k_t(s)$ in Eq. (4). **c** The two architectures of the VAN used in 2D (Methods). The left panels have a schematic of the gated recurrent unit cell (GRU)[57]. For 2D lattice systems, a zigzag path for transmitting variables is employed, allowing the hidden state to propagate both vertically and horizontally. The right panels are PixelCNNs. The vanilla PixelCNN[39] encounters an issue of the "blind spot", i.e., the black lattice point is not conditioned on the lattice points within the blind spot area under a 3 × 3 causal convolution layer. This issue is solved in the gated PixelCNN[40] by splitting the model into two parts: vertical and horizontal stacks.

## Algorithm

The pseudocode of tracking the dynamical partition function and dynamical phase transitions by the VAN is summarized below:

- **Input**: The system size and dimension, the model type, the boundary condition, time steps, values of the counting field $s$. Choose an initial distribution, such as the steady state of the non-tilted dynamics.
- Every time step $j = 0, \dots, J-1$:

1. Learn the next-step VAN $\hat{P}_{j+1}^{\theta_{j+1}}(\mathbf{x})$: for every epoch,
   (a) Draw samples $\{\mathbf{x}\}$ from $\hat{P}_{j+1}^{\theta_{j+1}}(\mathbf{x})$;
   (b) Calculate relevant matrix elements of the transition operator $\mathbb{T}_s$ to get $\mathbb{T}_s \hat{P}_j^{\theta_j}(\mathbf{x})$;
   (c) Train the VAN by minimizing the loss function: $\mathcal{L}_{j+1} = \mathbb{E}_{\mathbf{x} \sim P_{j+1}^{\theta_{j+1}}}[\ln \hat{P}_{j+1}^{\theta_{j+1}}(\mathbf{x}) - \ln \mathbb{T}_s \hat{P}_j^{\theta_j}(\mathbf{x})]$.
2. Calculate the variational free energy at each time step: $\mathcal{F}_{j+1}^{\theta_{j+1}}(s) = \mathcal{L}_{j+1}$ after training.

- Estimate the dynamical partition function: $Z_t(s) \approx \prod_{j=1}^{J} Z_j(s)$, $\ln Z_j(s) \approx \mathcal{F}_j^{\theta_j}(s)$; and moments of dynamical observables.
- **Output**: The evolved probability distributions, dynamical partition function, and dynamical observables.

The computational complexity of tracking the dynamical phase transition depends on the number of training steps $N_{\text{train}}$ and the number of counting-field values $N_s$. Under a fixed system size, the

computational time of studying the steady-state phase transition[34] has the order of $\mathcal{O}(N_s N_{\text{train}})$, and the finite-time study without the phase transition[25] has $\mathcal{O}(N_{\text{train}} J)$ for $J$ total time steps. Here, to uncover the phase transition at any time, the computational complexity becomes $\mathcal{O}(N_s N_{\text{train}} J)$, where larger $N_{\text{train}}$ may be required to reach high accuracy for larger system sizes (Supplementary Fig. 4). Despite the high computational complexity, the result under the chosen lattice sizes fulfills to uncover the finite-time scaling of the phase transition for 2D and 3D KCMs, as demonstrated below.

## Applications

The present approach is generally applicable to systems in none-equilibrium statistical mechanics. Since the computational cost is proportional to the number of allowable state transitions at each time step in Eq. (2), the cost is greatly reduced when compared with tracking the full distribution exponentially proportional to the system size. This is computationally feasible when the number of allowable transitions scales polynomially on the system size, including the voter model[6] which validates our method of tracking the distribution (Supplementary Fig. 1). Another representative class of systems is KCMs modeling glasses[8]. The KCMs exhibit rich phenomena of phase transitions (Fig. 2), where the 2D and 3D cases were unexplored either analytically or numerically, and are our focus.

**Active-inactive phase transition over time for KCMs.** The previous studies of KCMs focused on equilibrium properties[46] or the active-

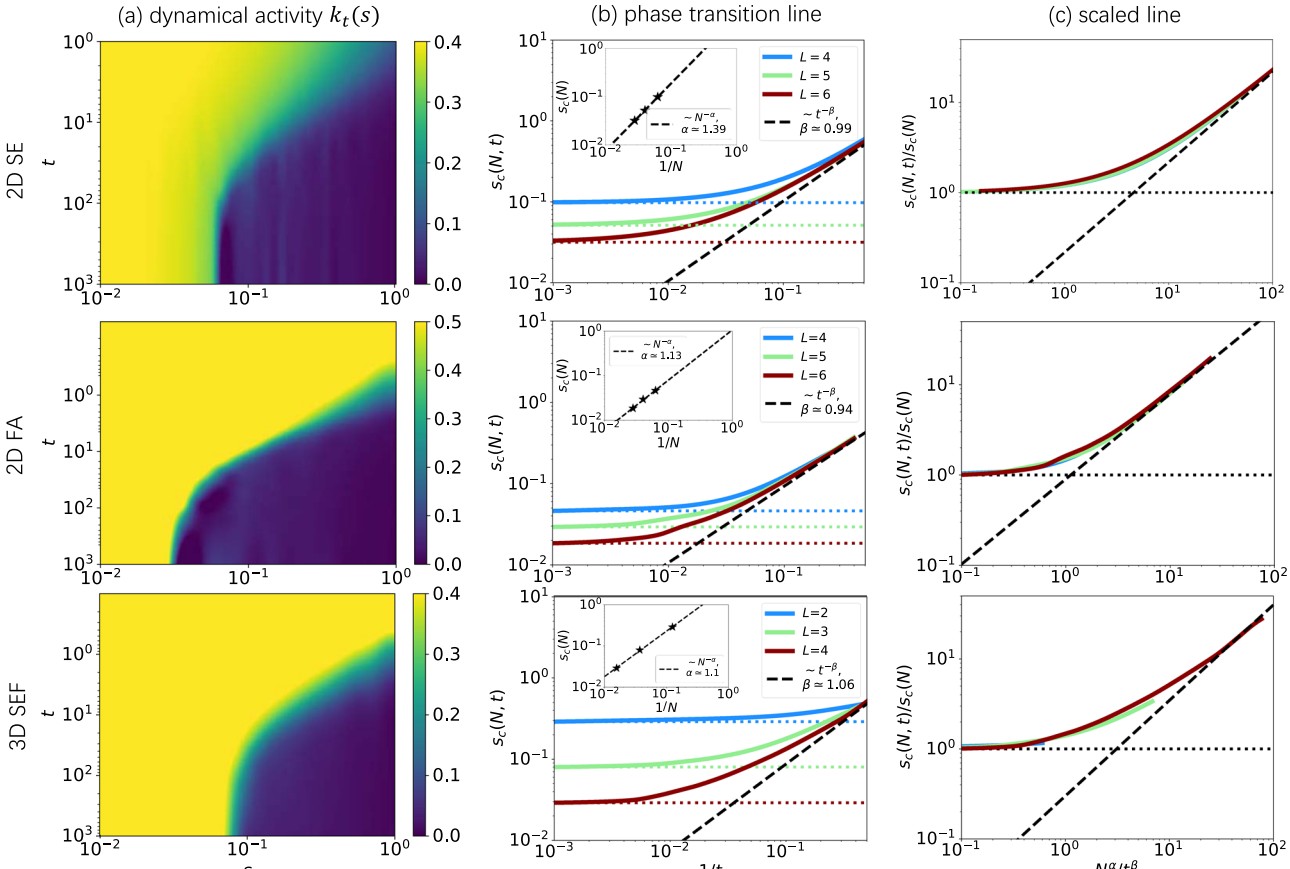

**Fig. 3 | Characterization of the dynamical active-inactive phase transition of kinetically constrained models.** The top, middle, and bottom panels are the 2D SE model, 2D FA model and 3D SEF model. **a** The dynamical activity $k_t(s)$ denoted by color reveals the phase transition versus time $t$ and the counting field $s$, with $L = 5$ for 2D and $L = 3$ for 3D. **b** The critical point $s_c(N, t)$ over time, giving critical exponents $\alpha$ from $s_c(N) \sim N^{-\alpha}$ at the steady state (the horizontal dotted lines and the inset) and $\beta$ from the finite-time scaling $t^{-\beta}$ (the black dashed line). **c** The scaled phase transition lines, with $s_c(N, t)/s_c(N)$ and time scaled as $N^\alpha t^{-\beta}$, are collapsed together, indicating the proper scaling relation. Parameters: $c = 0.5, L = 4, 5, 6$ for 2D and $L = 2, 3, 4$ for 3D.

inactive phase transition in the long-time limit[30–32,34]. For the phase transition at arbitrary time, besides the recent result in 1D[33], the cases with lattice dimension greater than 1 have not been revealed. Here, the VAN provides the phase transition over time of KCMs on 1D, 2D, and 3D lattices in a unified way.

We consider two paradigmatic KCMs, namely, FA[28] and variants of East[29] models, on a lattice of size $N = L$ in 1D, $N = L^2$ in 2D, and $N = L^3$ in 3D, with binary spins $x_i = 0, 1$ for $i = 1, \ldots, N$, $d_s = 2$ and $2^N$ configurations in total. The Markovian generator is

$$\mathbb{W} = \sum_{i=1}^{N} f_i [c\sigma_i^+ + (1-c)\sigma_i^- - c(1-x_i) - (1-c)x_i], \quad (10)$$

where $\sigma_i^\pm$ are the Pauli raising and lowering operators flipping site $i$ up and down, and $c \in (0, 0.5]$ controls the rate of flipping up. The up spin number $x_i$ acts as the operator $x_i = \sigma_i^+ \sigma_i^-$, and the terms $1 - x_i$, $x_i$ separately represent the escape transitions out from the down, up spins at site $i$. The $f_i$ equals the number of up spins at certain directions of the nearest neighbors (Fig. 2a): the FA model counts all directions, the 1D East counts the left, the 2D South-East (SE) counts the left and above, and the 3D South-East-Front (SEF) counts the left, above and back. We consider open boundary conditions for the convenience of comparing results with the literature. The boundary sites are up for 1D[33] and down for 2D[34] and 3D. For the 2D FA model, the configuration with all down spins is excluded to avoid this disconnected configuration. To access the largest ergodic element in the configuration space, the first spin is fixed up for 2D and 3D East

models. Note that all the figures of configurations do not include boundary sites.

The generator acts on each configuration and at each time step contributes to the case with only one spin flip between two configurations. Each configuration has $N$ connected configurations that transit into or out from. The probability distribution is updated by using only the connected configurations of batch samples in Eq. (8). This procedure reduces the complexity from multiplying the transition matrix with the probability vector, both of which are exponentially large, to counting an order of $\mathcal{O}(N)$ transitions linear to the system size.

The VAN shows a high accuracy in revealing the phase transition. The obtained dynamical partition function coincides with the numerically exact values available for the small system sizes (Supplementary Fig. 5). Our result at long time matches with the steady-state estimation from the variational Monte-Carlo method[34] (Supplementary Fig. 6). For the finite-time regime which was previously explored only for 1D, our result (Supplementary Figs. 2, 3) agrees with tensor networks[33]. For the unexplored 2D and 3D KCMs at finite time, we obtain the dynamical activity $k_t(s)$ as a function of time $t$ and the counting field $s$ with $s > 0$, showing two phases with extensive or subextensive activities (Fig. 3a). The critical point $s_c(N, t)$ as a function of system size and time can be identified numerically by the peak point of the dynamical susceptibility $\chi_t(s) = dk_t(s)/ds$. We conduct a scaling analysis of the critical point $s_c(N, t)$. In the long-time regime, the scaling of system size $s_c(N) \sim N^{-\alpha}$ gives the exponent $\alpha \gtrsim 1$ for 2D and 3D (Fig. 3b). By inspecting the short-time regime, the critical point approximately scales as $s_c(t) \sim t^{-1}$ for the two models. These two

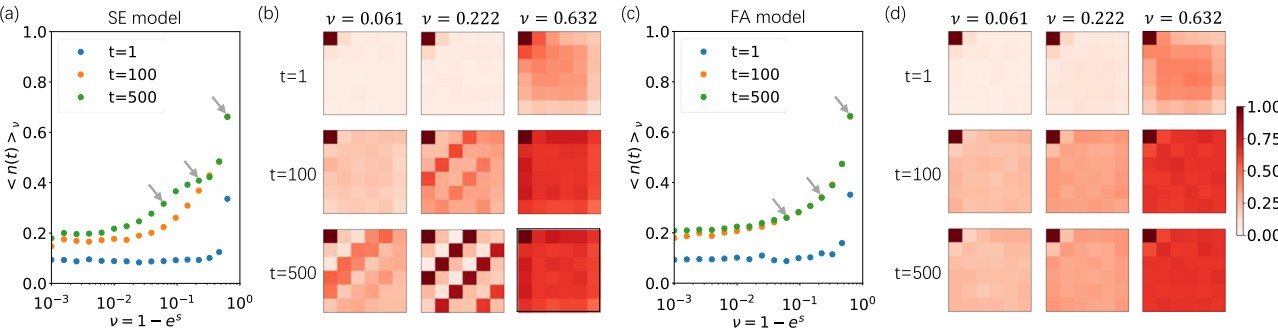

**Fig. 4 | The emergence of the active phases.** The 2D SE (**a**, **b**) and 2D FA (**c**, **d**) models with $c = 0.05$, $L = 6$ and $s < 0$. **a**, **c** The average density in the active phase at various time points for $c = 0.05$. The grey arrows point to the chosen $v$ (introduced for $s < 0$) for (**b**, **d**). **b**, **d** Time evolution of the spatial profile of configurations at various time points (rows) for different $v$ values (columns). For the comparison between the two models, the left-up corner spin of both models is fixed up. The color denotes the average number of up spins, with the same color bar in (**d**).

regimes motivate to approximate $s_c(N, t) \approx s_c(N) + s_c(t)$. We then estimate the scaling of time by fitting $\ln[s_c(N, t) - s_c(N)]$ versus $-\ln(t)$, giving $s_c(t) \sim t^{-\beta}$ with $\beta \approx 1$ for 2D and 3D, which confirms the speculated value. The finite-time scaling in 2D and 3D has a similar critical exponent as in 1D[33], implying that the transition point evolves with time similarly in all the dimensions for KCMs. By dividing $s_c(N, t)$ with $s_c(N)$ and time as $N^\alpha t^{-\beta}$, $s_c(N, t)$ curves are collapsed together (Fig. 3c).

**The emergence of active phases for 2D KCMs.** We further analyze the emergence of characteristic spatial structures over time, beyond the steady state[34]. For 2D SE and FA models, with a small $c = 0.05$, we discover that the average density of up spins $\langle n(t) \rangle_v$ (versus $v \equiv 1 - e^s$) of the 2D SE model gradually shows piecewise density plateaus (Fig. 4a), which are absent in the 2D FA model (Fig. 4c). For the average spatial profile of configurations, the 2D SE model forms structures with up-spin diagonal bands separated by down-spin bands, and the number of bands varies over $v$ (Fig. 4b). The 2D FA model does not have such characteristic bands (Fig. 4d), even when its first spin is also fixed up.

## Discussion

We have presented a general framework to track dynamics of nonequilibrium statistical systems based on neural networks, and developed an efficient algorithm to estimate the dynamical partition function. This extends applications of the VAN from equilibrium[19] to nonequilibrium, from the steady state[34] to arbitrary time, and from the absence of phase transitions[25] to phase transitions, the approach enables to reveal of the unexplored active-inactive phase transition at finite time and scaling relations in KCMs on 2D and 3D lattices, as well as the emergence of characteristic spatial structures.

The error, such as quantified by the relative error of the dynamical partition function in Eq. (16), does not keep increasing (Supplementary Figs. 4, 5), because the time evolution of the distribution does not have a dramatic change for all time points. The accumulation of error mainly occurs when the dynamics change dramatically over time, e.g., when the active-inactive phase transition occurs. Identifying these time points by trial and error helps find the most efficient way of increasing epochs for accuracy at certain time points. An alternative way of resolving the issue is to project the evolution into two parts: one follows the largest eigenvalue of the tilted generator, and the other follows from a modified tilted generator (Supplementary Sect. IIA). When the modified generator reaches a steady value, one can stop the simulation and extrapolate it by using the largest eigenvalue.

Capturing the active-inactive transition requires an efficient sampling of rare inactive configurations with few up spins, which is accessible by the importance sampling (Supplementary Sect. IIIA). This important sampling is on the configurations from a distribution at each time point, different from the sampling on trajectories, which may be

harder to sample as the trajectory space grows exponentially with time points. Besides, learning the multiple probability peaks can be affected by the mode-collapse: For rugged distributions, not all modes of the target distribution may be directly captured by the VAN[47]. To alleviate the mode-collapse, besides the importance sampling used here, temperature annealing[19] and variational annealing[48] can be employed.

When the system size increases, the computational time reaches the order of $\mathcal{O}(10^2)$ h for one value of the counting field (Supplementary Table 1). Larger system sizes require longer computational time to reach high accuracy (Supplementary Fig. 4). Evaluating the phase transition needs to scan various values of the counting field. Thus, although the method is generally applicable, computing the scaling relation for larger system sizes meets the practical challenges of available computational resources. However, this can be alleviated by optimizing the efficiency of tracking the distribution, with the help of the time-dependent variational method[49] and by paralleling multiple GPUs.

The present approach is applicable to other types of Markovian dynamics, including stochastic reaction networks[26,50], where phase transitions can be analyzed after adding the control parameter. Based on generality of the VAN, it is adaptable to other topologies, such as the voter model on graphs[51] and epidemic spreading on networks[52], where the architecture of the VAN can be the graph neural network[43]. It may also be generalized to the KCMs of various dimensions in the quantum regime[53]. Another direction is to leverage the VAN for sampling rare trajectories, with the help of the Doob operator[54], active learning[55] and reinforcement learning[56].

## Methods
### Variational autoregressive networks for spin-lattice systems
We use the variational autoregressive network[19] to parameterize the probability distribution. The VAN factorizes the joint probability into a product of conditional probabilities as Eq. (5), where $x_i$ denotes the $d_s$-state variable of site $i$ ($d_s = 2$ for binary spin systems). The symbol $\theta$ represents the learnable parameters. The parameterized distribution is automatically normalized, which is also called autoregressive modeling in the machine learning community. Since each conditional probability only depends on previous sites, it supports efficient ancestral sampling in parallel.

Below, we briefly describe the architecture of the RNN and gated PixelCNN for our problem. The setting of MADE was the same as[19].

**Recurrent neural networks.** For the recurrent neural network (RNN), we use a gated recurrent unit[57] as the recurrent cell, which is capable of learning the distribution with long-range correlations. It is more efficient than the long-short time memory (LTSM) model and avoids the vanishing gradient problem for vanilla recurrent neural networks.

For the 1D RNN, the conditional probability is iteratively obtained over the one-dimensional sites. A recurrent cell processes the information from the previous hidden state $h_{i-1}$ and the input data $x_{i-1}$ in the current cell, generates a new hidden state $h_i$, gives the conditional probability $\hat{P}^\theta(x_i|x_1, \ldots, x_{i-1})$ based on $h_i$, and passes on the information of $h_i$ to the next cell. The dimension of the hidden states is denoted by $d_h$. The GRU has a candidate hidden state $\hat{h}_i$, an update gate $z_i$ interpolating between the previous and candidate hidden states, and a reset gate $r_i$ setting the extent of forgetting for the previous hidden state. It updates by the following gates:

$$z_i = \sigma(W_{zx}x_{i-1} + W_{zh}h_{i-1} + b_z), \tag{11}$$

$$r_i = \sigma(W_{rx}x_{i-1} + W_{rh}h_{i-1} + b_r), \tag{12}$$

$$\hat{h}_i = \tanh(W_{hx}x_{i-1} + W_{hh}(r_i \odot h_{i-1}) + b_h), \tag{13}$$

$$h_i = (1 - z_i) \odot h_{i-1} + z_i \odot \hat{h}_i, \tag{14}$$

where $W$s are the weight matrices, $b$s are bias vectors, and $\sigma$ is the sigmoid activation function, $\odot$ denotes the Hadamard product.

The conditional probability is obtained from the hidden states. The output is acted on by a linear transform and a softmax operator $\hat{P}^\theta(x_i|x_1, \ldots, x_{i-1}) = \text{Softmax}(Wh_i + b)$, which ensures the normalized condition for the output probability vector. Given an initial hidden state $h_0$ and variable $x_0$ (chosen as a zero vector here), the full probability is obtained by Eq. (5) with the iteratively generated conditional probabilities. Sampling from the probability distribution is conducted similarly: given an initial hidden state and variable, the variable $x_1$ is sampled from the estimated conditional probability, and the procedure is repeated to the last site.

For the 2D RNN, the implementation is more involved. A zigzag path[34,41] is used to transmit the lattice variables, both vertically and horizontally. For the vertical and horizontal variables (hidden state $h$ and variable $x$ separately), we first concentrate the two into one and perform a linear transform (without the bias term) to an intermediate variable with the original dimension. They are then passed to the next GRU cell to continue the iteration.

**The gated PixelCNN.** For 2D lattice systems, we find that the vanilla PixelCNN model[39] suffers from the blind spot problem. In the worst case, the blind spot in the receptive field only covers half of the sites above and to the left of the current site. To circumvent the blind spot problem, we use the gated PixelCNN[40] that combines two convolutional network stacks: the vertical stack and the horizontal stack. The vertical stack conditions on all the sites left and the horizontal stack conditions on all the sites above. The activation function is also replaced by a gated activation unit:

$$\mathbf{h}^{l+1} = \tanh(W_1^l * \mathbf{h}^l) \odot \sigma(W_2^l * \mathbf{h}^l) \tag{15}$$

where $l$ is the number of layers, $\mathbf{h}^l$ is the feature map at the $l$-th layer, $\odot$ is the Hadamard product and $*$ denotes the stacked convolution operation.

### Details of training neural networks
When tracking the dynamics in Eq. (1), shorter Trotter time-step length generally gives higher accuracy, with the cost of longer simulation time. For KCMs, the range of time-step length $\delta t \in [0.01, 0.1]$ is found suitable, as also reported in the 1D case[33]. Considering both the accuracy and efficiency, $\delta t$ here is often chosen as $\delta t = 0.1$ for 1D and $\delta t = 0.05$ for 2D and 3D. The loss usually takes a number of $> \mathcal{O}(10^3)$ epochs to converge for the first time step of evolving the system but requires only an order of $\mathcal{O}(10^2)$ epochs for the following time steps, because the probability distribution only has a small change after each time step. The smaller number of epochs after the first time step saves the training time for tracking the evolution.

Learning rates affect the accuracy of training. Among the tested learning rates $10^{-5}, 10^{-4}, 10^{-3}$, and $10^{-2}$, we found that $10^{-3}$ typically leads to relatively lower loss values and better training. It is possible to encounter general optimization issues such as trapping into local minima, which may be alleviated by designing schedulers for the learning rate.

We used the Adam optimizer[58] to perform the stochastic gradient descent. The batch size was usually set as 1000 for each epoch. To better estimate the variational free energy, we used the batch from the last 20% of the training epochs, where the loss converges. This average gives a more accurate estimate by using approximately $1000 \times 100 \times 20\%$ (batch size, epochs at each time point, last percent of the training epochs) batch samples.

### The relative error
We estimate the error of the VAN for each system separately. To quantify the accuracy of the VAN, we calculated the relative error of the dynamical partition functions between the VAN and the numerically exact result. The numerically exact result of the dynamical partition function was obtained by summing up probabilities of all possible states, which is feasible only for systems with small sizes, e.g., approximately $L \le 10$ for 1D, $L \le 4$ for 2D, and $L \le 2$ for 3D.

The relative error $e_r$ is defined as:

$$e_r = \left| \frac{\ln Z_t(s) - \ln Z_t(s)_{\text{exact}}}{\ln Z_t(s)_{\text{exact}}} \right|, \tag{16}$$

where $\ln Z_t(s)_{\text{exact}}$ is the numerically exact result by summing up probabilities of all possible states, which is feasible for systems with small sizes. The error is shown in the inset of Supplementary Fig. 2a for 1D and Supplementary Fig. 5 for 2D and 3D. All the figures show that the relative error is usually smaller than $\mathcal{O}(10^{-3})$ compared with the numerically exact result, validating the accuracy of the VAN. Based on the accuracy and efficiency, we chose the more appropriate VAN for each dimension: the RNN in 1D, the gated PixelCNN in 2D, and the MADE in 3D.

For larger system sizes, the numerically exact result is not accessible which then demands the use of the present algorithm. Thus, we can only evaluate the error from the loss function Eq. (7) based on the KL-divergence: the lower value of the loss function implies more accurate training, with the lower bound given by the dynamical partition function. We remark that the loss function will not reach zero, even when the VAN is accurately learned because the probability is not normalized under the tilted generator. Under this case, it is not feasible to estimate the KL divergence of the two probability distributions at consecutive time points as a quantification of accuracy. Besides, the value of the loss function decreases with a larger number of epochs and increases with the system size (Supplementary Fig. 4), indicating the requirement for more epochs and longer computational time when the system size increases.

## Data availability
The authors declare that the data supporting this study are available within the paper.

## Code availability
A PyTorch implementation of the present algorithm can be found in Supplementary Data 1 and at the GitHub repository (https://github.com/Machine-learning-and-complex-systems/DPT).

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

## Acknowledgements

We thank Juan P. Garrahan, Luke Causer, and Corneel Casert for their helpful communication. We also acknowledge Online Club Nanothermodynamica for discussions. This work is supported by Project 11747601 (P.Z.), 12325501 (P.Z.), 12247104 (P.Z.), 12322501 (Y.T.), and 12047503 of the National Natural Science Foundation of China. P.Z. is partially supported by the Innovation Program for Quantum Science and Technology project 2021ZD0301900 and ZDRW-XX-2022-3-02 of the Chinese Academy of Sciences. The high-performance computing is supported by Dawning Information Industry Corporation Ltd.

## Author contributions

P.Z., Y.T. had the original idea for this work. Y.T., J.L. and J.Z. performed the study, and all authors contributed to the preparation of the manuscript.

## Competing interests

The authors declare no competing interests.
