## [Peer Review File · Nature Communications]

REVIEWER COMMENTS

Reviewer #1 (Remarks to the Author):

This article focuses on dynamical phase transitions of kinetically constrained models. Initiated more than a decade ago, numerous methods have been proposed to address this problem. The primary reason for this is the necessity to employ "tilted" dynamics to study dynamical phase transitions. In this dynamics, probability is not conserved, preventing researchers from using a standard technique like MCMC. Rare event sampling methods, such as transition path sampling or population dynamics algorithm, were initially used to tackle this problem. However, these methods were later replaced by more model-specific techniques, such as those involving matrix product states or tensor networks. Still, studying dynamical phase transitions in finite time was not possible. This article demonstrates that by using a neural network model - specifically, the Variational Autoregressive Network (VAN) - this problem can be comprehensively studied. I am impressed by these results and am inclined to support the publication of this article.

Technically, there are a few points I'd like the authors to address:

1. The VAN model assumes a specific expression for the probability distribution: Eq.(5). Given that no mathematical proofs confirm that Eq.(5) is exact for KCM problems, there needs to be some form of justification for this assumption's validity. In this article, the authors use the following strategy to justify it:

(a) They apply the VAN model and obtain results for parameters (such as smaller system size, long time, and lower dimension) where other methods can also be applied. These results are then compared, confirming that the VAN model is functional within this regime.

(b) Subsequently, they apply the VAN model to a finite-time case, an area where the solution remains unknown.

While I agree that this strategy is valid, I wonder if a better method might exist. As Eq.(5) is crucial to the application of the VAN model, I recommend the authors to devote at least a paragraph to discussing this critical issue.

2. Regarding the computational time, the authors discussed computational complexity in the discussion section, where they mention "larger N_{train} may be required for larger system sizes.". I think it's important to clarify the dependency of N_{train} on system size. For example, does N_{train} increase exponentially or linearly as the system size increases? Similarly, the authors compared the results from the VAN model with other methods in an infinite time setting and in a

lower dimension. I'd recommend the authors also compare the computational costs associated with obtaining these results.

3. The authors unveiled phase diagram for 2D and 3D KCMs at finite time. These results are entirely new, and I'm genuinely impressed. However, I would like the authors to look deeper into the physical implications of their finding. Demonstrating that dynamical phase transitions are not merely a mathematical exercise, but also reflect real-world physics, is of crucial importance.

Below are minor comments and corrections for typos:

- On p. 3, just below eq.(4), "in a similar was" should be corrected to "in a similar way".
- On p. 6, at the end of the paragraph, it's written "Boundary conditions are in Supplementary Information". If I understand correctly, the boundary condition is periodic. If so, I would recommend the authors to specify it in the main text.

Reviewer #2 (Remarks to the Author):

The manuscript titled "Learning Nonequilibrium Statistical Mechanics and Dynamical Phase Transitions" introduces an innovative approach. Utilizing autoregressive neural networks, this method enables simulation of nonequilibrium statistical mechanics and facilitates the exploration of the physics underlying dynamical phase transitions.

This algorithm, premised on autoregressive neural networks, is powerful. It possesses the distinct capability to tackle high-dimensional systems, exceeding the boundaries achievable by existing methodologies. The algorithm's presentation is clear, supported by comprehensive supplementary materials and source data.

The authors have successfully employed this approach to probe the active-inactive phase transition over time for KCMs and to explore the active phases for 2D KCMs. The 2D and 3D results, including data collapse and critical exponent extraction, are both interesting and valuable to the community, given that the previous methods are usually limited to 1D systems.

This work is a nice illustration of how machine learning can contribute to the advancement of the study of nonequilibrium statistical mechanics. However, before recommending publication in Nature Communications, there are several comments and it will be helpful to hear back from the authors.

Comments

The algorithm is presented in this work is closely related to the algorithm published in Nature Machine Intelligence volume 5, pages 376–385 (2023). Can the authors comment a bit more what is new from this work given the previous algorithm? Does the new feature come from the fact that the probability after each step may not be normalized? It will be helpful to elaborate a bit more in the manuscript for the readers' information.

The authors have written several complexity discussion of the algorithm in the discussion. It may be more helpful to have them earlier in the Results section after the VAN algorithm is presented, since this could provide more information for the readers to understand. In addition, even though the algorithm could work in high dimension, there could be accumulated error in each step optimization. Is it possible for the authors to comment a bit on how the error of this approach in terms of system size and the evolution time?

For the Figure.2, the authors use $L=4$ for 3D on the data collapse. Can the authors comment a bit why $L=4$ is sufficient, or how difficult to get to larger L on 3D?

One of the nice contribution of the work is to explore the physics of nonequilibrium statistical physics, especially in 2D and 3D. However, it seems that the current discussion of the physics is a bit short and it could be more detailed. For example,

i) the authors mention “FA model counts all directions (Fig. 2a), the 1D East counts the left, the 2D South-East (SE) counts the left and above, and the 3D South-East-Front (SEF)”. It may not be too clear for some readers that are not in the field to have an intuitive picture of how these models look like. Is it possible for the authors to provide more elaboration or cartoons on those models for understanding?

ii) in Figure.2(b), the authors provide some cartoon on activate-inactivate phase transition. Can the authors elaborate a bit more why these are called activate or inactivate phase from seeing the cartoon so that the general readers can gain more intuition? In addition, is there an easy way to explain how these cartoon reflect the change of $kt(s)$?

iii) the authors mention “with a small $c = 0.05$, we discover that the average density of up spins $\langle n(t) \rangle_v$ (versus $v \equiv 1 - es$) of the 2D SE model gradually shows density plateaus (Fig. 3a), which are

absent in the 2D FA model (Fig. 3c).” According to Fig. 3(a), the green dot shows certain plateau behavior around $v \sim 10^{-1}$, but later seems to go higher as v increases. Could the authors explain whether this is a true plateau?

iv) Following the point (iii), the authors highlight the difference between SE model and FA model. Is it possible to provide a physics theory or picture to explain why these two models have the different phenomena? In addition, it seems that both SE model and FA model both approach more similar density pattern for $v = 0.632$, can the authors help provide more elaboration?

I'm glad to reconsider a revised version.

Reviewer #3 (Remarks to the Author):

In the manuscript "Learning nonequilibrium statistical mechanics and dynamical phase transitions", the authors developed a computational framework using variational autoregressive networks to study the time evolution of nonequilibrium systems.

The goal is to address the lack of methods for time evolution beyond the steady state and for higher dimensions.

The authors apply this approach to typical constraint models (like the east or the voter model) of nonequilibrium systems, discovering the active-inactive phase transition of spin flips, uncovering a dynamical phase diagram, and identifying new scaling relations.

We recommend accepting the paper with minor revisions, suggestions are detailed below.

Are they novel and do you think that they represent a significant advance in the field? If the conclusions are not original, it would be helpful if you could provide relevant references.

The manuscript presents a novel technique for investigating time-evolution in nonequilibrium systems. By applying this approach, the authors study for example the phase transition of the East model (2D, 3D) at various time points, revealing crucial insights into the dynamical phase diagram concerning time, the control parameter of the counting field, and the critical exponent of the finite-time scaling. The authors state that this has not been obtained either analytically or numerically previously.

In this sense, the results are indeed novel and allow the study of dynamical phenomena in models that were previously difficult to investigate. Previous studies like (10.1103/PhysRevX.10.021051) focused on one-dimensional systems.

Is the work convincing, and if not, what further evidence would be required to strengthen the conclusions?

In its current state, the method is generally convincing given the impressive numerical results but would benefit from a wider discussion of errors, scaling, and limitations.

A) When studying the time evolution of quantum mechanical systems (unitary evolution of pure states), we generally see an accumulation of errors with increasing size of the Trotter time steps and duration of the evolution. As we understand it, this aspect is either not observed with this method or it is not discussed in the text. Please add a more detailed discussion of the limitations in terms of maximal time reached and the influence of the approximations (like Trotterization).

B) The presented results are shown for relatively small system sizes ($L = 4, 5, 6$ for 2D and $L = 2, 3, 4$ for 3D), hence a discussion on how scaling to larger systems affects the performance should be included. In the discussion (page 7) the scaling is linear in the number of training samples (N_{train}), the number of counting field values (N_s) and the maximal number of time steps (J). In terms of complexity, this is an impressive scaling, however, it is unclear how the parameters relate to the system size. Is it possible that there is a hidden dependence? The limitation to small systems suggests so. It would be beneficial for the manuscript to give a more detailed evaluation here, especially since Nature Communications is oriented towards a wider audience.

C) Given that the training times in the Supplementary Information are already in the range of hours, a high prefactor could be expected to the scalings given above.

The manuscript would benefit from including a discussion on the limitations of the proposed method. The errors cited in IV.C are rather small. Would it be possible to push the method further and if not, what are the limitations? Training time, accuracy, etc?

By addressing these aspects, the work would become more convincing and provide a well-rounded perspective on the applicability and potential constraints of this novel approach.

On a more subjective note, do you feel that the paper will influence thinking in the field and be of interest to a wider community of researchers?

The paper presents an interesting use case of neural network based machine learning models to track the normalized joint distribution of configurations to study dynamical phenomena. The developed tool could become interesting for a wider community of researchers if a proper discussion of its limitations is included.

If you recommend publication, please outline, in a paragraph or so, what you consider to be the outstanding features.

Before accepting the manuscript for publication, we would suggest to the authors addressing a few points regarding the discussion and presentation, which are explained in the comments. Nonetheless, we would be looking forward to seeing an improved and revised version published.

##General Comments

1. For the discussion of errors, the manuscript currently merely states that an “error of $O(10^{-3})$ compared to the numerically exact result” is achieved; we believe a more substantiated discussion on how errors influence the quality of the evolved dynamical partition function is needed. As stated earlier, we generally see an accumulation of errors with increasing time steps and duration of the evolution. How does this relate to the method presented in this manuscript?

2. In the main text of the manuscript there are extensive references to Supplementary Information, especially in the beginning of the results section (page 6, Applications, II B (1)). This makes some parts of the work difficult to read, which could be fixed by adding 1-2 figures into the main text. As far as we know, the maximal number of figures in Nature Communications is not exceeded yet (<https://www.nature.com/ncomms/submit/article#:~:text=Figure legends are limited to,figures and/or tables>). Furthermore, some of the references to the Supplementary Information are very unpecific. Please add more details on what section in the Supplementary Information is referred to.

3. Page 3: “Eqs. (1) (2)” -> “and” is missing.

4. Page 4: “Typical neural-network architectures are employed, including MADE [38], PixelCNN [39] or RNN [40]. The VAN can be further improved by cooperating with more advanced neural-network architectures and sampling techniques.”

Please be more specific concerning which architecture is used for which particular instance, otherwise the sentence poses more questions than it answers.

5. Page 5: The inset in Fig2cii is quite difficult to read because of its size, maybe there is an alternative way of presenting the information.

6. Page 6: "Boundary conditions are in Supplementary Information." -> Boundary conditions are explained in the Supplementary Information (Section?).
7. Page 6: Some of the figures suffer from an overlap in axis ticks (see Fig3a/c). We understand the drive to minimize space consumption figures. Please do not compromise on readability.
8. Page 7: "Despite the high computational complexity" -> The complexity is linear in all given quantities (see above). There seems to be a confusion between computational complexity and actual runtime. An algorithm might be expensive in runtime, although it scales well. Are there hidden dependencies on the system size here that influence the scaling? (see comment A above)
9. Page 7: "The accumulation of the error" -> Please provide more details on which errors in particular.
10. Page 8: Section on Recurrent neural networks, explanation of GRU: The explanation is well written but difficult to understand without a figure. Our suggestion is to insert a figure similar to Fig1c here.
11. Page 8: "blind spot problem" -> Please include some details on what the blind spot problem is. Nature Communications is aimed at a rather broad audience and not everyone is necessarily familiar with the blind spot problem.
12. Page 9: "The less required epoch after the first time step" -> Reformulation to: "The smaller number of epochs after ..."? Would this be equivalent?
13. Page 9: Section on Details of training neural networks: In our opinion, this section includes too many details on the research story, which could be included in the supplementary information. Please be concise and set the focus on the actual configuration used to obtain the results.
14. Page 9: "The imposed symmetry does not significantly improve the accuracy of the training or the estimation on the dynamical partition function, and requires longer computational time." -> Please discuss this point a bit more in detail as it is counterintuitive. Symmetries should reduce the number of configurations and the size of the Hilbert space.

15. Page 9: “We estimated the error of the VAN under each dimension” -> The wording is slightly unclear. Is “We estimate the error of the VAN for each system separately.” equivalent?

16. Page 9: “ $O(10^{-3})$ compared with the numerically exact result, validating the accuracy of the VAN.” -> This part should include discussion on limitations and challenges. (See comment C above)

Comments on the supplementary material

17. Same concern as in comment 7: Some of the figures suffer from an overlap in axis ticks (see Fig3).

Comments on the provided code

We tested the provided code to the degree that we ran the files Main1D.py, Main2D.py, Main3D.py, MainVoter.py. After installing the necessary packages, all files (except for one, see comment 22) ran without problems. We did not check the correctness of numerical results.

18. To enhance the quality and lower the hurdle to reuse the provided code, incorporating comprehensive documentation would be beneficial. This could include docstrings (<https://peps.python.org/pep-0257/>) and documentation within the respective functions.

19. The readability of the provided code can be significantly improved by removing the substantial amount of unused (#) code.

20. It would be beneficial to translate a few comments from Chinese to English (see the 'gru.py' file).

21. To alleviate the burden to install all Python packages separately, please include a requirements.txt file that details all packages needed and their version. You can export the current package configuration from your virtual environment with `pip freeze`.

22. Testing to run “MainVoter.py” fails with “UnboundLocalError: local variable 'Col1' referenced before assignment”, please test this file again to remove the error.

Reviewer #4 (Remarks to the Author):

Responses to Reviewers - NCOMMS-23-13809A-Z

We thank all the reviewers for their careful reading and constructive comments, which led to a better manuscript. We are glad that all the reviewers are generally positive and suggest minor revisions:

- Reviewer 1 found the manuscript impressive and comprehensive;
- Reviewer 2 recognized the present approach as innovative and the algorithm’s presentation clear;
- Reviewer 3 co-reviewed with reviewer 4 found that the manuscript presents a novel technique and impressive numerical results.

In response to the reviewers’ comments and suggestions, we have carefully revised the manuscript. An overview of the changes is given below, followed by point-to-point responses to reviewers’ comments. Original reports are typed in orange color, our response in black, and the revised text in blue. Please note the numbers of equations and figures refer to the revised manuscript. The quoted references are listed at the end of the response.

List of main modifications in the revised manuscript

1. Figures:

- Fig. 2 and Fig. 3 have been reorganized and revised.
- Fig. 2 now contains an illustrative panel on KCMs and dynamical phases, as well as the schematic representation of GRU and gated PixelCNN based on the reviewer’s suggestion.
- Fig. 3 now includes numerical results for 2D and 3D KCMs enlarged for better visualization.
- Fig. 4 has been revised to avoid the overlap of the axis ticks.

2. Results section:

- We elaborated on the effectiveness of the VAN in representing the distribution in nonequilibrium statistical mechanics.
- We specified the neural-network architectures employed for KCMs.
- In the caption of Algorithm 1, we demonstrated the significant difference between the present algorithm for a tilted generator and our previous work on the chemical master equation with a non-tilted generator.
- For clarity, we moved the setting of boundary conditions from Supplementary Information to the section “Application”.

- We discussed the physical implication of the new finite-time scaling in the phase diagram of 2D and 3D KCMs.
 - We explained in more details that the density plateau depicted in Fig. 4 exhibits a piecewise characteristic.
 - We refined the presentation of specific sentences according to the reviewers' suggestions.
3. Discussion section:
- We specified that the mentioned accumulation of the error is quantified by Eq. (16).
 - We elaborated on the issue of computational cost, with respect to the system size.
 - We discussed the limitation on studying larger-size systems and strategies for further improvements.
4. Methods section:
- We analyzed the accuracy and error accumulation through the loss function.
 - Following reviewers' suggestions, we moved the technical details of the neural network from the Methods section to Supplementary Information.
5. In references
- We added references such as [1] for KCMs in the quantum regime.
6. Supplementary Information:
- Supplementary Fig. 4 has been added to demonstrate the evaluation of the accuracy and its dependence on the number of training epochs and the system size.
 - In Supplementary Fig. 7, we added the computational cost of the numerically exact result and the steady-state calculation from the variational Monte-Carlo method.
 - The supplementary figures have been revised to avoid the overlap of axis ticks.
 - In Sec. VB, we provided details on quantifying the accuracy and its dependence on the number of epochs as well as system sizes.

Response to Reviewer 1

This article focuses on dynamical phase transitions of kinetically constrained models. Initiated more than a decade ago, numerous methods have been proposed to address this problem. The primary reason for this is the necessity to employ “tilted” dynamics to study dynamical phase transitions. In this dynamics, probability is not conserved, preventing researchers from using a standard technique like MCMC. Rare event sampling methods, such as transition path sampling or population dynamics algorithm, were initially used to tackle this problem. However, these methods were later replaced by more model-specific techniques, such as those involving matrix product states or tensor networks. Still, studying dynamical phase transitions in finite time was not possible. This article demonstrates that by using a neural network model - specifically, the Variational Autoregressive Network (VAN) - this problem can be comprehensively studied. I am impressed by these results and am inclined to support the publication of this article.

Response: We are grateful for the reviewer’s meticulous review of our manuscript and appreciate the positive feedback, particularly noting the comprehensiveness and impressive aspects of our work.

Technically, there are a few points I’d like the authors to address:

1. The VAN model assumes a specific expression for the probability distribution: Eq. (5). Given that no mathematical proofs confirm that Eq. (5) is exact for KCM problems, there needs to be some form of justification for this assumption’s validity. In this article, the authors use the following strategy to justify it:
 - (a) They apply the VAN model and obtain results for parameters (such as smaller system size, long time, and lower dimension) where other methods can also be applied. These results are then compared, confirming that the VAN model is functional within this regime.
 - (b) Subsequently, they apply the VAN model to a finite-time case, an area where the solution remains unknown. While I agree that this strategy is valid, I wonder if a better method might exist. As Eq. (5) is crucial to the application of the VAN model, I recommend the authors to devote at least a paragraph to discussing this critical issue.

Response: We agree with the reviewer. As the mathematical formulation in Eq. (5), the VAN factorizes the joint distribution as $P(\mathbf{x}) = \prod_{i=1}^N P(x_i|x_1, \dots, x_{i-1})$. In *autoregressive modeling* of machine learning, the conditionals are parameterized by neural networks with trainable parameters θ , which was found to have a strong representative power on the joint distribution in many applications. For example, it has been applied in equilibrium statistical mechanics [2], and quantum physics [3]. It was also shown to be effective in the study on the steady state of 2D KCMs [4]. Based on the evidence and the results presented in our manuscript, we believe Eq. (5) is a natural variational *ansatz* for parameterizing the joint distribution of KCMs at any finite time.

Accordingly, we added a paragraph in the revised manuscript to discuss the justification

of Eq. (5): “The VAN is capable of expressing strongly correlated distributions [5, 6], including equilibrium distributions in statistical mechanics [2], steady state distributions of KCMs in nonequilibrium statistical mechanics [4] and quantum systems [3]. Here, we find it effective to learn the time-evolving distributions. The expressivity of the VAN and training time depends on the architecture, as well as the depth and width of the neural network (Supplementary Sect. VD). Typical neural-network architectures can be employed, including MADE [7], PixelCNN [8, 9] or RNN [10]. For example, we have used RNN for KCMs in 1D and 2D, PixelCNN for 2D, and MADE for 3D.”

For the reviewer’s question on whether a better method for justifying the VAN model exists, we provide two approaches to quantify the accuracy, as detailed in the response to the next point below.

2. Regarding the computational time, the authors discussed computational complexity in the discussion section, where they mention “larger N_{train} may be required for larger system sizes.”. I think it’s important to clarify the dependency of N_{train} on system size. For example, does N_{train} increase exponentially or linearly as the system size increases?

Response: We thank the reviewer for mentioning the crucial point. To demonstrate how the accuracy depends on N_{train} and the system size, we first need to have a measure on the accuracy, for which there are two approaches: (1) comparing with other methods, and (2) using an intrinsic estimator such as the loss function. To examine the dependency of N_{train} on system size, we need to deal with system sizes such as $L = 5, 6$ in 2D, where other methods are not available. Under the case, we use the loss function to quantify the accuracy. In general, the lower value of the loss function implies more accurate training. Then, we change N_{train} , for example, from 100 epoch to 1, 10 epoch at each time point. The result shows that smaller N_{train} gives larger loss (Supplementary Fig. 4b). This is repeated for different system sizes, where loss increases almost linearly with the system size (Supplementary Fig. 4c).

Here we have mainly considered a fixed largest N_{train} to evaluate the relative accuracy of smaller N_{train} under different system sizes, due to the following reason. We notice that the loss function here is not the KL-divergence of two probability distributions, because the probability is no longer normalized under the tilted generator. The loss function will not reach zero even when the VAN is faithfully learnt, and has the lower bound given by the dynamical partition function. Due to this technical issue of a lack of precise quantification on the accuracy, it is not straightforward to conclude whether N_{train} depends on exponentially or linearly with the system size. Consequently, we have used the strategy in the last paragraph. These issues are now discussed in Supplementary Sect. VB.

3. Similarly, the authors compared the results from the VAN model with other methods in an infinite time setting and in a lower dimension. I’d recommend the authors also compare the computational costs associated with obtaining these results.

Response: Thanks for the suggestion. In Fig. 7 of the revised manuscript, we provide the computational time of other numerical methods, such as the variational Monte-Carlo method for the steady state and the numerically exact method of calculating the probability of all configurations. Although the numerically exact method has a relatively short computational time under the chosen system size, it is only feasible for small system sizes as the computational complexity increases exponentially in the dimension.

4. The authors unveiled phase diagram for 2D and 3D KCMs at finite time. These results are entirely new, and I'm genuinely impressed. However, I would like the authors to look deeper into the physical implications of their finding. Demonstrating that dynamical phase transitions are not merely a mathematical exercise, but also reflect real-world physics, is of crucial importance.

Response: Following the reviewer's suggestion, we added a more detailed discussion on the physical implications. First, the phase transition of spin flips gives insights into the dynamical heterogeneity of the glass transition. As discussed in [11], the picture of dynamic heterogeneity corresponds to that glassiness is not necessarily a consequence of static interactions, but of effective constraints on the dynamics [12]. The simplest microscopic model for illustrating this is KCMs.

Second, the KCMs have a dynamical transition between active and inactive phases, depending on whether the spins were able to make the surrounding spins flip [13]. In general, the location of the transition point depends both on time and counting field (similar to the temperature in equilibrium statistical physics), however, the finite-time scaling of active-inactive phase transitions has not been possible until the recent one-dimensional result [14]. With the present method, we are able to provide the dynamical phase diagram and the finite-time scaling for 2D and 3D cases. Especially, we find that the finite-time scaling in 2D and 3D are similar to that in 1D, showing that the transition point shifts with time similarly in all dimensions for these models.

We have added the discussions to the revised manuscript.

5. Below are minor comments and corrections for typos:
 - On p. 3, just below eq.(4), "in a similar was" should be corrected to "in a similar way".

Response: Thanks. We have fixed it in the revised manuscript.

- On p. 6, at the end of the paragraph, it's written "Boundary conditions are in Supplementary Information". If I understand correctly, the boundary condition is periodic. If so, I would recommend the authors to specify it in the main text.

Response: Thanks for the suggestion. We now have specified boundary conditions in the main text. For the convenience of comparing results with the literature, we consider open boundary conditions, where the boundary sites are up for 1D

[14] and down for 2D [4] and 3D. For the 2D FA model, the configuration with all down spins is excluded to avoid this disconnected configuration, as the setup in [4]. We also remark that the present neural network method works equally well with a periodic boundary condition because the neural network representation of the probability distribution does not rely on the boundary condition.

Finally, we thank the reviewer again for the constructive suggestions.

Response to Reviewer 2

The manuscript titled “Learning Nonequilibrium Statistical Mechanics and Dynamical Phase Transitions” introduces an innovative approach. Utilizing autoregressive neural networks, this method enables simulation of nonequilibrium statistical mechanics and facilitates the exploration of the physics underlying dynamical phase transitions.

This algorithm, premised on autoregressive neural networks, is powerful. It possesses the distinct capability to tackle high-dimensional systems, exceeding the boundaries achievable by existing methodologies. The algorithm’s presentation is clear, supported by comprehensive supplementary materials and source data.

The authors have successfully employed this approach to probe the active-inactive phase transition over time for KCMs and to explore the active phases for 2D KCMs. The 2D and 3D results, including data collapse and critical exponent extraction, are both interesting and valuable to the community, given that the previous methods are usually limited to 1D systems.

This work is a nice illustration of how machine learning can contribute to the advancement of the study of nonequilibrium statistical mechanics. However, before recommending publication in Nature Communications, there are several comments and it will be helpful to hear back from the authors.

Response: We thank the reviewer for the careful reading and positive assessment of the manuscript. Following the reviewer’s comments below, we have added more detailed discussions and carefully revised the manuscript.

1. The algorithm is presented in this work is closely related to the algorithm published in Nature Machine Intelligence volume 5, pages 376–385 (2023). Can the authors comment a bit more what is new from this work given the previous algorithm? Does the new feature come from the fact that the probability after each step may not be normalized? It will be helpful to elaborate a bit more in the manuscript for the readers’ information.

Response: Thanks for the great suggestion, which helps clarify our method. As noticed by the reviewer, a major difference is that the dynamical operator (tilted generator) does not conserve the probability, under which the evolved probability vector becomes unnormalized, such that the conventional approach developed in [15] does not apply. To solve the issue of unnormalized probability, we developed a new algorithm that can track the time evolution under the tilted generator. Another important feature of the current work is that it enables computing the dynamical partition function and then studying the phase transition. It is achieved by the renormalization procedure, where minimizing the loss function gives the normalization constant at each time point when the VAN faithfully learns the evolved distribution. The renormalization procedure over time points is absent in [15]. To make the new aspect more transparent for readers, we have clarified this point in the manuscript:

“The renormalization procedure over time points enables to extract the dynamical partition function under the tilted generator, beyond the algorithm of only tracking the evolving distribution under the non-tilted generator [15].”

In the legend of the ALGORITHM 1, we add: “Compared with the algorithm that tracks the time evolution under the non-tilted generator [15], the present algorithm is capable of dealing with the tilted generator, under which the evolved probability vector becomes unnormalized. This enables to calculate the dynamical partition function, a central quantity to study the phase transition.”

2. The authors have written several complexity discussion of the algorithm in the discussion. It may be more helpful to have them earlier in the Results section after the VAN algorithm is presented, since this could provide more information for the readers to understand. In addition, even though the algorithm could work in high dimension, there could be accumulated error in each step optimization. Is it possible for the authors to comment a bit on how the error of this approach in terms of system size and the evolution time?

Response: Following the reviewer’s suggestion, we have moved the discussion of the computational complexity to the Results section, right after the VAN algorithm is presented. The more detailed analysis and discussions are in Methods and Supplementary Information.

To evaluate how the accuracy depends on time, there are two approaches: (1) comparing with other methods, and (2) using an intrinsic estimator. For (1), we now have compared the evolving distribution from the VAN with the numerically exact result for small system sizes. The KL-divergence between distributions of the two methods is generally small (Supplementary Fig. 4a), and mainly increases when the system transits from the active phase to the inactive phase, where the error can be further reduced by using more training epochs near the transition time or employing the strategies in Supplementary Sect. III. Besides, the relative error of the dynamical partition function is typically below the order of $\mathcal{O}(10^{-3})$ (Supplementary Fig. 5), demonstrating that the training does not suffer from a serious error accumulation over time.

For larger system sizes, such as $L = 5, 6$ in 2D, no other methods are available. Thus, the only currently available way to evaluate the error over time is by (2): the lower value of the loss function generally implies more accurate training. Now, we have shown the loss function to demonstrate the accuracy of learning the distribution (Supplementary Fig. 4b) over time points, at the count field with the phase transition. The result demonstrates that the loss function will decrease when the system evolves towards the steady state, without a serious error accumulation.

We also examine how the accuracy depends on the system size. Specifically, we evaluate the change of the time-averaged loss function with different system sizes. The result shows that the loss function, under the same number of training epochs, increases

almost linearly with the system size (Supplementary Fig. 4c), indicating that the error does not dramatically increase with the system sizes. However, for even larger system sizes, the computational time may further increase (see the response to the next point).

3. For the Figure.2, the authors use $L=4$ for 3D on the data collapse. Can the authors comment a bit why $L=4$ is sufficient, or how difficult to get to larger L on 3D?

Response: Thanks for the question. We would like to mention that here we mainly use $L = 2, 3, 4$ to demonstrate that the method is capable of revealing the phase diagram for 3D: having three lattice sizes already enables us to obtain the scaling relation. Second, the present method is generally applicable to nonequilibrium statistical systems, however, in practice sufficient accuracy requires more computational time. For example, the total number of spins of $L = 5$ in 3D will be 125. Given the computational resources at hand, it becomes more challenging to reach enough accuracy, for example, the memory of GPU A100 may not be enough.

To reach high accuracy, the training may also need more epochs at each time point (Supplementary Fig. 4). For system sizes larger than the chosen ones, the currently used 100 epoch of training steps may no longer be accurate enough. according to the computational time in Supplementary Fig. 7, larger L ($L > 4$ in 3D) would require the scale of $\mathcal{O}(10^2)$ hours for one value of the counting field. To fully evaluate the phase transition line the critical exponent needs to scan various values of the counting field, which may become unaffordable under the currently available computational resources. Thus, although the method is generally applicable, computing the scaling relation for larger system sizes in 3D meets the practical challenges. We anticipate that the issue can be alleviated by optimizing the efficiency of tracking the distribution and by paralleling multiple GPUs in future studies.

4. One of the nice contribution of the work is to explore the physics of nonequilibrium statistical physics, especially in 2D and 3D. However, it seems that the current discussion of physics is a bit short and it could be more detailed. For example,

- (i) the authors mention “FA model counts all directions (Fig. 2a), the 1D East counts the left, the 2D South-East (SE) counts the left and above, and the 3D South-East-Front (SEF)”. It may not be too clear for some readers that are not in the field to have an intuitive picture of how these models look like. Is it possible for the authors to provide more elaboration or cartoons on those models for understanding?

Response: We thank the reviewer for the great suggestion. We have revised Fig. 2 to give a physical intuition about the models. Specifically, in Fig. 2a, we have provided an illustrative picture of the flipping rules of the two-dimensional kinetically constrained models, i.e., the 2D FA model and the 2D SE model. The 3D SEF model has a similar rule as the 2D SE model, with the counting of neighbors in 3D. This clarification should have provided an intuitive understanding

of the dynamics of these models.

- (ii) in Figure 2(b), the authors provide some cartoons on activate-inactivate phase transition. Can the authors elaborate a bit more why these are called activate or inactivate phases from seeing the cartoon so that the general readers can gain more intuition? In addition, is there an easy way to explain how these cartoon reflect the change of $k_t(s)$?

Response: Following the reviewer’s comment, we now have revised the figure to better illustrate the active and inactive phases, by adding the rephrase allocated to the two phases separately in Fig. 2b. The activate or inactivate phase corresponds to the case where the spin flips frequently or infrequently. We have also explained the connection to the change of the dynamical activity $k_t(s)$ in the figure.

- (iii) the authors mention “with a small $c = 0.05$, we discover that the average density of up spins $\langle n(t) \rangle_\nu$ (versus $\nu \equiv 1 - e^s$) of the 2D SE model gradually shows density plateaus (Fig. 3a), which are absent in the 2D FA model (Fig. 3c).” According to Fig. 3(a), the green dot shows certain plateau behavior around $\nu \sim 10^{-1}$, but later seems to go higher as ν increases. Could the authors explain whether this is a true plateaus?

Response: Thanks for the question. We first would like to clarify that the density plateau is piecewise, as first proven for the steady state in 1D [16] and later numerically observed in 1D [17] and 2D [4]. Here, for the 2D SE model at long times, the plateau behavior appears around $\nu \sim 10^{-1}$ and then disappears when ν further increases. Correspondingly, the spatial profile of configurations forms an anticorrelated structure with up-spin diagonal bands separated by down-spin bands. Such behaviors are absent from the FA model. We remark that for both the models, the average density is spatially featureless at short times, consistent with the 1D finite-time result [14]. As time increases, the average-up density gradually increases, forming piecewise density plateaus only for the 2D SE model, which is also similar to the 1D case [14].

- (iv) Following the point (iii), the authors highlight the difference between SE model and FA model. Is it possible to provide a physics theory or picture to explain why these two models have the different phenomena? In addition, it seems that both SE model and FA model both approach more similar density pattern for $\nu = 0.632$, can the authors help provide more elaboration?

Response: Following the above response, for ν values with density plateaus, the corresponding configurations in the 2D SE model form diagonal bands of up spins surrounded by vacant bands. This behavior is different from the 2D FA model, where density plateaus are absent and spatial profiles are homogeneous. The physical origin of the different behaviors of the two models may be explained by their distinct microscopic interactions. The SE model has only the interactions from

the south and east sides, which leads to spatially anti-correlated structures along the diagonal direction. Instead, the FA model counts all the nearest neighbors' interactions, leading to more homogeneous spatial profiles.

For larger ν such as $\nu = 0.632$, the dynamics tend to be more active and the configurations with more upper spins dominate. This happens for both the two models, where their configurations have uniformly more upper spins. That is, the $\nu = 0.632$ is out of the range of the density plateau, such that configurations in the SE model no longer have the characteristic density pattern.

Finally, we thank the reviewer again for the constructive comments and thoughtful suggestions.

Response to Reviewer 3

In the manuscript "Learning nonequilibrium statistical mechanics and dynamical phase transitions", the authors developed a computational framework using variational autoregressive networks to study the time evolution of nonequilibrium systems. The goal is to address the lack of methods for time evolution beyond the steady state and for higher dimensions. The authors apply this approach to typical constraint models (like the east or the voter model) of nonequilibrium systems, discovering the active-inactive phase transition of spin flips, uncovering a dynamical phase diagram, and identifying new scaling relations.

We recommend accepting the paper with minor revisions, suggestions are detailed below.

Response: We thank the reviewer for the careful reading of our manuscript. We are also grateful for the reviewers' pithy summary of the major results and the recommendation on the acceptance after minor revisions. Below, we have carefully addressed the reviewers' questions and revised the manuscript accordingly.

1. The manuscript presents a novel technique for investigating time evolution in nonequilibrium systems. By applying this approach, the authors study for example the phase transition of the East model (2D, 3D) at various time points, revealing crucial insights into the dynamical phase diagram concerning time, the control parameter of the counting field, and the critical exponent of the finite-time scaling. The authors state that this has not been obtained either analytically or numerically previously. In this sense, the results are indeed novel and allow the study of dynamical phenomena in models that were previously difficult to investigate. Previous studies like (10.1103/PhysRevX.10.021051) focused on one-dimensional systems.

Response:

We thank the reviewer for bringing out the relevant literature. We now have mentioned the possible generalization of the present method to the quantum system, in the Discussion of the revised manuscript: "It may also be generalized to the KCMs of various dimensions in the quantum regime [1]."

2. In its current state, the method is generally convincing given the impressive numerical results but would benefit from a wider discussion of errors, scaling, and limitations.
 - (A) When studying the time evolution of quantum mechanical systems (unitary evolution of pure states), we generally see an accumulation of errors with increasing size of the Trotter time steps and duration of the evolution. As we understand it, this aspect is either not observed with this method or it is not discussed in the text. Please add a more detailed discussion of the limitations in terms of maximal time reached and the influence of the approximations (like Trotterization).

Response: Thanks for the great advice. We first would like to mention that the error does depend on the size of the Trotter time steps, similar to that in quantum systems. In general, smaller sizes of the time step give higher accuracy,

with the cost of longer simulation time. For KCMs, the range $\delta t \in [0.01, 0.1]$ was found typically suitable [14]. Thus, the time step length here is chosen as $\delta t = 0.1$ for 1D and $\delta t = 0.05$ for 2D and 3D, which gives a sufficiently low relative error compared with the numerically exact result (Supplementary Fig. 5). We now have discussed the effect of time-step length in Methods:

“When tracking the dynamics in Eq. (1), shorter Trotter time-step length generally gives higher accuracy, with the cost of longer simulation time. For KCMs, the range of time-step length $\delta t \in [0.01, 0.1]$ is found suitable, as also reported in the 1D case [14]. Considering both the accuracy and efficiency, δt here is often chosen as $\delta t = 0.1$ for 1D and $\delta t = 0.05$ for 2D and 3D.”

Second, although in principle the error should generally accumulate over time steps, in the examples here, we observe that the error does not always keep increasing, such as when comparing the evolving distribution with the numerically exact result (Supplementary Figs. 4a). The error mainly increases when the system transits from the active phase to the inactive phase, where the error can be further reduced by using more training epochs near the transition time or employing the strategies in Supplementary Sect. III. Besides, the relative error of the dynamical partition function also shows that its time evolution does not have a dramatic change for all time points (Supplementary Figs. 5). For larger system sizes when other methods are not available, we now also show the loss function, which does not significantly increase over time (Supplementary Fig. 4b), indicating that the accumulation of error is not serious in KCMs. Especially, when the time evolution is close to the steady state, the error will no longer increase. For this reason, the recent 1D finite-time study [14] proposed that the partition function can be directly extrapolated once the system reaches the steady state.

- (B) The presented results are shown for relatively small system sizes ($L = 4, 5, 6$ for 2D and $L = 2, 3, 4$ for 3D), hence a discussion on how scaling to larger systems affects the performance should be included. In the discussion (page 7) the scaling is linear in the number of training samples (N_{train}), the number of counting field values (N_s) and the maximal number of time steps (J). In terms of complexity, this is an impressive scaling, however, it is unclear how the parameters relate to the system size. Is it possible that there is a hidden dependence? The limitation to small systems suggests so. It would be beneficial for the manuscript to give a more detailed evaluation here, especially since Nature Communications is oriented towards a wider audience.

Response: We thank the reviewers for the comments. To address the question, we first need a measure on the accuracy of the training. For small system sizes, this has been evaluated by comparing with the numerically exact result. For larger system sizes, e.g., $L = 5, 6$ in 2D, no other methods are available. Thus, we have compared with the steady-state variational Monte-Carle method, to ensure the

accuracy in the long-time regime (Supplementary Fig. 6). For the finite-time regime, since the present method is the first achievable for 2D and 3D, the only currently available way to evaluate the accuracy is by the loss function.

To demonstrate how the accuracy depends on the system size, we now use the following strategy. For one value of s with a relatively large loss value, we change N_{train} , for example, from 100 epoch to 1, 10 epoch at each time point, and evaluate the change of loss with N_{train} . This is repeated for different system sizes, providing the dependence of the accuracy on system size. The result shows that a smaller epoch does give a larger loss (Supplementary Fig. 4b) and the loss increases with the system size (Supplementary Fig. 4c). Consequently, for even larger system sizes, it may require > 100 epoch to have the loss sufficiently small. Thus, although the method is generally applicable, computing the scaling relation for larger system sizes meets the practical challenges. However, this can be alleviated such as by paralleling multiple GPUs. Moreover, the result under the chosen lattice sizes already fulfills to uncover the finite-time scaling of the phase transition for 2D and 3D KCMs.

We have added these points in Discussion and Supplementary Sect. VB, and mentioned the dependence of N_{train} on system sizes in the Results section: “Here, to uncover the phase transition at any time, the computational complexity becomes $\mathcal{O}(N_s N_{train} J)$, where larger N_{train} may be required to reach high accuracy for larger system sizes (Supplementary Fig. 4).”

- (C) Given that the training times in the Supplementary Information are already in the range of hours, a high prefactor could be expected to the scalings given above. The manuscript would benefit from including a discussion on the limitations of the proposed method. The errors cited in IV.C are rather small. Would it be possible to push the method further and if not, what are the limitations? Training time, accuracy, etc?

By addressing these aspects, the work would become more convincing and provide a well-rounded perspective on the applicability and potential constraints of this novel approach.

Response: Following the reviewer’s comment, we now have provided potential constraints due to the computational time in the main text. As the reviewers mentioned, the computational time to reach high accuracy will increase for larger system size (Supplementary Fig. 7), which is the order of $\mathcal{O}(10^2)$ hours and becomes unaffordable. Evaluating the phase transition needs to further scan various values of the counting field. Thus, computing scaling relation would take weeks or months, and has not been conducted due to the limit of our computational resources. This is the practical limitation of the present method, even though the method is generally applicable. We plan to further optimize the efficiency of our code and then systematically study larger system sizes as future studies.

Accordingly, we now have added the discussion in the main text: “When the system size increases, the computational time reaches the order of $\mathcal{O}(10^2)$ hours for one value of the counting field (Supplementary Fig. 7). Larger system sizes require longer computational time to reach high accuracy (Supplementary Fig. 4). Evaluating the phase transition needs to scan various values of the counting field. Thus, although the method is generally applicable, computing the scaling relation for larger system sizes meets the practical challenges of available computational resources. However, this can be alleviated by optimizing the efficiency of tracking the distribution, with the help of the time-dependent variational method [18] and by paralleling multiple GPUs.”

3. The paper presents an interesting use case of neural network based machine learning models to track the normalized joint distribution of configurations to study dynamical phenomena. The developed tool could become interesting for a wider community of researchers if a proper discussion of its limitations is included.

Response: We appreciate the positive comments. We would like to mention that the present result not only achieved the mentioned point but also enabled tracking of the evolution of probabilities under a non-conserved dynamical operator and obtaining the dynamical partition function. Now, we have also included a discussion on the limitations of the method as mentioned above.

4. Before accepting the manuscript for publication, we would suggest that the authors address a few points regarding the discussion and presentation, which are explained in the comments. Nonetheless, we are looking forward to seeing an improved and revised version published.

Response: We are grateful to the reviewer’s encouraging comment and insightful comments for the revision.

General Comments

1. For the discussion of errors, the manuscript currently merely states that an “error of $\mathcal{O}(10^{-3})$ compared to the numerically exact result” is achieved; we believe a more substantiated discussion on how errors influence the quality of the evolved dynamical partition function is needed. As stated earlier, we generally see an accumulation of errors with increasing time steps and duration of the evolution. How does this relate to the method presented in this manuscript?

Response: Thanks for the question. The error of the estimated distribution (Supplementary Figs. 4a) and dynamical partition function (Supplementary Figs. 5) do not keep increasing, because the time evolution of the distribution may not have a dramatic change for all time points. For example, when the time evolution is close to the steady state, the error will no longer increase. Once the system reaches a steady state, the value of the partition function was proposed to be directly extrapolated [14]. This supports that the error accumulation over time is not serious. We have elaborated on

this point in the Discussion section:

“The error, such as quantified by the relative error of the dynamical partition function in Eq. (16), does not keep increasing (Supplementary Figs. 4, 5), because the time evolution of the distribution does not have a dramatic change for all time points. The accumulation of error mainly occurs when the dynamics change dramatically over time, e.g., when the active-inactive phase transition occurs. Identifying these time points by trial and error helps find the most efficient way of increasing epochs for accuracy at certain time points.”

2. In the main text of the manuscript there are extensive references to Supplementary Information, especially at the beginning of the results section (page 6, Applications, II B (1)). This makes some parts of the work difficult to read, which could be fixed by adding 1-2 figures into the main text. As far as we know, the maximal number of figures in Nature Communications is not exceeded yet (<https://www.nature.com/ncomms/submit/article#:~:text=Figurelegendsarelimitedto,figuresand/or%20tables>). Furthermore, some of the references to the Supplementary Information are very unspecific. Please add more details on what section in the Supplementary Information is referred to.

Response: Thanks for the suggestion. First, we have extended the previous Fig. 2 to be the new Fig. 2 and Fig. 3, for better illustrations. For the figures in the Supplementary Information, we find that they are mainly about the details of the result for supporting the main figures. Also, after the revision, the main text almost reaches the length limit. Thus, we would like to keep Supplementary figures in Supplementary.

Following the reviewers' suggestion, we now have annotated specifically which section or figure of the Supplementary Information is quoted in the main text.

3. Page 3: “Eqs. (1) (2)” -> “and” is missing.

Response: Thanks for pointing this out. We have fixed it in the revised manuscript.

4. Page 4: “Typical neural-network architectures are employed, including MADE [38], PixelCNN [39] or RNN [40]. The VAN can be further improved by cooperating with more advanced neural-network architectures and sampling techniques.” Please be more specific concerning which architecture is used for which particular instance, otherwise the sentence poses more questions than it answers.

Response: Thanks for the suggestion. Now, the choice of specific neural network architecture for KCMs is provided here. The more detailed settings, such as the depth and width of the neural networks, are given in the Methods section. In the main text, it reads: “Typical neural-network architectures can be employed, including MADE [7], PixelCNN [8] or RNN [10]: for example, we have used RNN for KCMs in 1D and 2D, PixelCNN for 2D, and MADE for 3D. In 1D, we find RNN more accurate than MADE. In 2D, RNN has comparable accuracy with PixelCNN but takes a longer computational time (Methods).”

5. Page 5: The inset in Fig2cii is quite difficult to read because of its size, maybe there is an alternative way of presenting the information.

Response: We appreciate the reviewer’s thoughtful suggestion and have revised the figure, which is the current Fig. 3. Now, the font size of the inset should be sufficiently large to be visualized. We prefer to leave the size-scaling relation as inset figures because it can be obtained from the variational Monte-Carlo method [4].

6. Page 6:” Boundary conditions are in Supplementary Information.” -> Boundary conditions are explained in the Supplementary Information (Section?).

Response: Thanks for the suggestion. The boundary conditions are now given in the section of Applications in the main text: “We consider open boundary conditions for the convenience of comparing results with the literature. The boundary sites are up for 1D [14] and down for 2D [4] and 3D. For the 2D FA model, the configuration with all down spins is excluded to avoid this disconnected configuration.”

7. Page 6: Some of the figures suffer from an overlap in axis ticks (see Fig3a/c). We understand the drive to minimize space consumption figures. Please do not compromise on readability.

Response: We agree with the reviewer’s suggestion. We have revised Fig. 4 (the previous Fig. 3) to be less condensed, avoiding the overlap of axis ticks.

8. Page 7: “Despite the high computational complexity” -> The complexity is linear in all given quantities (see above). There seems to be a confusion between computational complexity and actual runtime. An algorithm might be expensive in runtime, although it scales well. Are there hidden dependencies on the system size here that influence the scaling? (see comment A above)

Response: We have discussed the computational complexity and its dependence on the system sizes more thoroughly, as in the response to Comment A above. In particular, the computational time depends on the system size: to maintain similar accuracy, more training steps may be required for larger systems. We have added Supplementary Fig. 4 for analyzing the dependence of computational time on the system sizes. For the computational time of the chosen system sizes, we now have provided more details and listed them in Supplementary Fig. 7.

9. Page 7: “The accumulation of the error” -> Please provide more details on which errors in particular.

Response: We now have provided the detail of the mentioned error: “The error, such as quantified by the relative error in Eq. (16),...”

In Eq. (16), the error is defined as:

$$e_r = \left| \frac{\ln Z_t(s) - \ln Z_t(s)_{\text{exact}}}{\ln Z_t(s)_{\text{exact}}} \right|,$$

where $\ln Z_t(s)_{\text{exact}}$ is the numerically exact result by summing up probabilities of all possible states, which is feasible for systems with small sizes.

10. Page 8: Section on Recurrent neural networks, explanation of GRU: The explanation is well written but difficult to understand without a figure. Our suggestion is to insert a figure similar to Fig1c here.

Response: Thanks for your suggestion. We add a schematic representation of the GRU in Fig. 2c of the main text. The left two panels include the architecture of the GRU unit and the composition of GRU units for the 2D lattice systems.

11. Page 8: “blind spot problem” -> Please include some details on what the blind spot problem is. Nature Communications is aimed at a rather broad audience and not everyone is necessarily familiar with the blind spot problem.

Response: Thanks for your suggestion. We have also added a schematic representation of the blind spot problem in Fig. 2c of the main text. The right two panels about the PixelCNNs illustrate the blind spot problem and the solution by the gated PixelCNN, as proposed in the literature [9].

12. Page 9: “The less required epoch after the first time step” -> Reformulation to: “The smaller number of epochs after ...”? Would this be equivalent?

Response: We agree with the reviewer on the revision and have reformulated it to “The smaller number of epochs after the first time step”.

13. Page 9: Section on Details of training neural networks: In our opinion, this section includes too many details on the research story, which could be included in the supplementary information. Please be concise and set the focus on the actual configuration used to obtain the results.

Response: Based on the reviewer’s suggestion, we have moved some details of training neural networks, such as the choice of depth and width, to Supplementary Sect. VD, “Computational details for kinetically constrained models”. Now, the section “Details of training neural networks” in Methods contains concise information about the training detail.

14. Page 9: “The imposed symmetry does not significantly improve the accuracy of the training or the estimation on the dynamical partition function, and requires longer computational time.” -> Please discuss this point a bit more in detail as it is counter-intuitive. Symmetries should reduce the number of configurations and the size of the Hilbert space.

Response: Below we show a result after adding symmetry, indicating that symmetry does not improve the accuracy. The way of imposing the symmetry here is to perform corresponding symmetric transformations (e.g., the rotational plus reflectional symmetry for 2D cases) on the configurations sampled from the VAN, and then add these

configurations into the calculation, as also demonstrated in Appendix D of [10].

(a) The dynamical partition function and relative error compared with the numerically exact result, for the 2D SE model with (a) or without (b) imposing the symmetry. The computational time for (a) is around 2.01 hours, and for (b) is 4.83 hours. The same number of training epochs is used for the two cases. The lattice size is $L = 4$.

The result for the 2D SE model in the figure here shows that imposing the symmetry does not improve the accuracy of calculating the dynamical partition function. Indeed, the case with imposing symmetry has a high relative error, especially in the early-time regime, which may be due to that longer training is needed to reach higher accuracy. Overall, for the problem under study, imposing symmetry does not provide an additional advantage. We finally note that imposing the symmetry is an option in our code package, applied by an argument “z2” with the implementations in gru2D.py, such that the readers can further look into the results conveniently.

15. “We estimated the error of the VAN under each dimension” -> The wording is slightly unclear. Is “We estimate the error of the VAN for each system separately.” equivalent?

Response: Thanks for your great suggestion, which provides a clearer presentation. We have fixed it in the revised manuscript.

16. Page 9: “ $\mathcal{O}(10^{-3})$ compared with the numerically exact result, validating the accuracy of the VAN.” -> This part should include a discussion on limitations and challenges. (See comment C above)

Response: We agree with the reviewer. As the response to comment C above, we now have discussed the limitations and challenges, by adding a paragraph in the Discussion section. In the Methods section on Page 10, we have also added a more detailed discussion on the quantification of the accuracy:

“For larger system sizes, the numerically exact result is not accessible which then demands the use of the present algorithm. Thus, we can only evaluate the error from the loss function Eq. (7) based on the KL divergence: the lower value of the loss function

implies more accurate training, with the lower bound given by the dynamical partition function. We remark that the loss function will not reach zero, even when the VAN is accurately learned because the probability is not normalized under the tilted generator. Under this case, it is not feasible to estimate the KL divergence of the two probability distributions at consecutive time points as a quantification of accuracy. Besides, the value of the loss function decreases with a larger number of epochs and increases with the system size (Supplementary Fig. 4), indicating the requirement for more epochs and longer computational time when the system size increases.”

Comments on the supplementary material

17. Same concern as in comment 7: Some of the figures suffer from an overlap in axis ticks (see Fig3).

Response: Thanks for pointing this out. The overlap of axis ticks in Fig. 3 has been fixed now. The overlap of the supplementary figures has also been revised, such as Supplementary Fig. 2 and Supplementary Fig. 3.

Comments on the provided code

We tested the provided code to the degree that we ran the files Main1D.py, Main2D.py, Main3D.py, MainVoter.py. After installing the necessary packages, all files (except for one, see comment 22) ran without problems. We did not check the correctness of numerical results.

Response: We thank the reviewer for the very careful examination of our code. We are glad that almost all files ran without problems.

18. To enhance the quality and lower the hurdle to reuse the provided code, incorporating comprehensive documentation would be beneficial. This could include docstrings (<https://peps.python.org/pep-0257/>) and documentation within the respective functions.

Response: We appreciate the suggestion and we have incorporated docstrings into the code to enhance its readability.

19. The readability of the provided code can be significantly improved by removing the substantial amount of unused (#) code.

Response: Thanks for your suggestion. We have removed extraneous commentaries in the code. There are some commented codes that can be useful for Windows users, such that they can directly run the script by uncommenting the code. Thus, we have kept those commented code.

20. It would be beneficial to translate a few comments from Chinese to English (see the ‘gru.py’ file).

Response: Sure, we have translated the Chinese comments within the code into English, including the gru.py file.

21. To alleviate the burden to install all Python packages separately, please include a requirements.txt file that details all packages needed and their version. You can export the current package configuration from your virtual environment with ‘pip freeze‘.

Response: Thanks. We have incorporated README.md and requirements.txt files within the codebase. The Markdown document therein provides comprehensive instruction on configuring the environment for the purpose of reproducing the results of our manuscript.

22. Testing to run “MainVoter.py” fails with “UnboundLocalError: local variable ‘Coll’ referenced before assignment”, please test this file again to remove the error.

Response: We apologize for the confusion. We have revised the script “MainVoter.py” to avoid the error, and have tested it on the server and personal laptop.

Finally, we thank the reviewer again for the careful reading and constructive suggestions.

Response to Reviewer 4

Response: We thank the reviewer for carefully co-reviewing the manuscript. In the response to the reviewer 3 above, we have carefully addressed the reviewers' questions.

References

- [1] Nicola Pancotti, Giacomo Giudice, J. Ignacio Cirac, Juan P. Garrahan, and Mari Carmen Bañuls. Quantum east model: Localization, nonthermal eigenstates, and slow dynamics. *Phys. Rev. X*, 10:021051, Jun 2020.
- [2] Dian Wu, Lei Wang, and Pan Zhang. Solving statistical mechanics using variational autoregressive networks. *Phys. Rev. Lett.*, 122:080602, Feb 2019.
- [3] Giuseppe Carleo, Ignacio Cirac, Kyle Cranmer, Laurent Daudet, Maria Schuld, Naftali Tishby, Leslie Vogt-Maranto, and Lenka Zdeborová. Machine learning and the physical sciences. *Rev. Mod. Phys.*, 91:045002, Dec 2019.
- [4] Corneel Casert, Tom Vieijra, Stephen Whitelam, and Isaac Tamblyn. Dynamical large deviations of two-dimensional kinetically constrained models using a neural-network state ansatz. *Phys. Rev. Lett.*, 127:120602, Sep 2021.
- [5] Pankaj Mehta, Marin Bukov, Ching-Hao Wang, Alexandre GR Day, Clint Richardson, Charles K Fisher, and David J Schwab. A high-bias, low-variance introduction to machine learning for physicists. *Phys. Rep.*, 2019.
- [6] B. McNaughton, M. V. Milošević, A. Perali, and S. Pilati. Boosting monte carlo simulations of spin glasses using autoregressive neural networks. *Phys. Rev. E*, 101:053312, May 2020.
- [7] Mathieu Germain, Karol Gregor, Iain Murray, and Hugo Larochelle. Made: Masked autoencoder for distribution estimation. *arXiv:1502.03509*, 2015.
- [8] Aaron Van Oord, Nal Kalchbrenner, and Koray Kavukcuoglu. Pixel recurrent neural networks. In *International conference on machine learning*, pages 1747–1756. PMLR, 2016.
- [9] Aaron Van den Oord, Nal Kalchbrenner, Lasse Espeholt, Oriol Vinyals, Alex Graves, et al. Conditional image generation with pixelcnn decoders. *Advances in neural information processing systems*, 29, 2016.
- [10] Mohamed Hibat-Allah, Martin Ganahl, Lauren E. Hayward, Roger G. Melko, and Juan Carrasquilla. Recurrent neural network wave functions. *Phys. Rev. Research*, 2:023358, Jun 2020.
- [11] Juan P. Garrahan and David Chandler. Geometrical explanation and scaling of dynamical heterogeneities in glass forming systems. *Phys. Rev. Lett.*, 89:035704, Jul 2002.
- [12] R. G. Palmer, D. L. Stein, E. Abrahams, and P. W. Anderson. Models of hierarchically constrained dynamics for glassy relaxation. *Phys. Rev. Lett.*, 53:958–961, Sep 1984.
- [13] F. Ritort and P. Sollich. Glassy dynamics of kinetically constrained models. *Adv. Phys.*, 52(4):219–342, 2003.

- [14] Luke Causer, Mari Carmen Bañuls, and Juan P. Garrahan. Finite time large deviations via matrix product states. *Phys. Rev. Lett.*, 128:090605, Mar 2022.
- [15] Ying Tang, Jiayu Weng, and Pan Zhang. Neural-network solutions to stochastic reaction networks. *Nat. Mach. Intell.*, 5(1):376–385, 2023.
- [16] Robert L Jack and Peter Sollich. Large deviations of the dynamical activity in the east model: analysing structure in biased trajectories. *J. Phys. A*, 47(1):015003, dec 2013.
- [17] Mari Carmen Bañuls and Juan P. Garrahan. Using matrix product states to study the dynamical large deviations of kinetically constrained models. *Phys. Rev. Lett.*, 123:200601, Nov 2019.
- [18] Moritz Reh, Markus Schmitt, and Martin Gärttner. Time-dependent variational principle for open quantum systems with artificial neural networks. *Phys. Rev. Lett.*, 127:230501, Dec 2021.

REVIEWERS' COMMENTS

Reviewer #1 (Remarks to the Author):

We appreciate all the modifications made by the authors. Based on their revisions, I'm happy to recommend the publication of their article in Nature Communications.

Reviewer #2 (Remarks to the Author):

Thanks the authors for the update and I have gone through the all responses. The revised paper has addressed all my questions and made a nice contribution to the field. I would like to recommend it to be published.

Reviewer #3 (Remarks to the Author):

We would like to extend our appreciation for the effort the authors put into revising the manuscript in response to the concerns and suggestions raised during the peer review process. We are pleased to report that the revisions have enhanced the quality and clarity of the work for us.

The extensive response to the feedback provided and the improvements made in the revised manuscript lead us to recommend to accept this work.

In particular, we appreciate the additional analysis on scaling the method to larger system sizes (Supplementary Fig. 6), updated figure 2, and additional discussion on errors.

Reviewer #4 (Remarks to the Author):
